# Rapid memory encoding in a recurrent network model with behavioral time scale synaptic plasticity

**Pan Ye Li**, **Alex Roxin***

Centre de Recerca Matemàtica, Barcelona, Spain

* aroxin@crm.cat

**Data Availability Statement:** The authors confirm that all data underlying the findings are fully available without restriction. The numerical code and simulated data are available at https://github.com/PanYe87/BTSP-Based-Learning.

## Abstract

Episodic memories are formed after a single exposure to novel stimuli. The plasticity mechanisms underlying such fast learning still remain largely unknown. Recently, it was shown that cells in area CA1 of the hippocampus of mice could form or shift their place fields after a single traversal of a virtual linear track. In-vivo intracellular recordings in CA1 cells revealed that previously silent inputs from CA3 could be switched on when they occurred within a few seconds of a dendritic plateau potential (PP) in the post-synaptic cell, a phenomenon dubbed Behavioral Time-scale Plasticity (BTSP). A recently developed computational framework for BTSP in which the dynamics of synaptic traces related to the pre-synaptic activity and post-synaptic PP are explicitly modelled, can account for experimental findings. Here we show that this model of plasticity can be further simplified to a 1D map which describes changes to the synaptic weights after a single trial. We use a temporally symmetric version of this map to study the storage of a large number of spatial memories in a recurrent network, such as CA3. Specifically, the simplicity of the map allows us to calculate the correlation of the synaptic weight matrix with any given past environment analytically. We show that the calculated memory trace can be used to predict the emergence and stability of bump attractors in a high dimensional neural network model endowed with BTSP.

## Author summary

A recently discovered form of in-vivo plasticity, called Behavioral Time-scale Plasticity (BTSP), leads to the generation of a place cell in CA1 from a previously silent cell after a single intracellular plateau potential (PP). We show that this one-shot learning process is well-described by a 1D map, which updates the synaptic weight matrix after the PP. We use the map to study the storage of spatial memories in a recurrent network, such as CA3. The map allows us to calculate the correlation of the weight matrix with all past explored environments analytically. When only a small fraction of cells are place cells in any given environment, we show that the full dynamics of a recurrent network endowed with BTSP is equivalent to its projection onto a series of weakly interacting manifolds, one for each environment. Interestingly, this weak interaction, which takes the form of quenched

**Funding:** This work was funded by a grant from the Spanish Ministry of Science and Innovation PID2021-124702OB-I00 (AR). This work is supported by the Spanish State Research Agency, through the Severo Ochoa and Maria de Maeztu program for Centers and Units of Excellence in R&D (CEX2020- 001084-M) (AR). We thank CERCA Program/Generalitat de Catalunya for institutional support. The funders had no role in study design, data collection and analysis, decision to publish, or preparation of the manuscript.

**Competing interests:** The authors have declared that no competing interests exist.

variability in the network connectivity, actually enhances the memory capacity by stabilizing remote memories which otherwise would be unretrievable.

## Introduction

The physiological mechanisms leading to the formation of episodic memories are still largely unknown, despite decades of clinical, experimental, and theoretical work. The fact that episodes are, by definition, encoded in a one-shot fashion, poses a particular challenge to theories of Hebbian learning which rely on repeated pre-post pairings of neuronal activity to drive changes in synaptic efficacy [1–3]. However, recent findings on CA1 place cell dynamics in awake behaving mice provide insight into a potential synaptic mechanism underlying one-shot learning. Indeed, place cells in CA1 can appear spontaneously as mice explore a novel environment [4–7]. Intracellular recordings of CA1 activity in-vivo in head-fixed mice have revealed that the emergence of place activity is dependent on the occurrence of a plateau potential (PP) in the cell [5, 6]. Such PPs can occur as a consequence of input from the entorhinal cortex, which generates a calcium spike in the apical dendrite, but can also be induced through current injection at the soma. In either case, from the resultant tuned response of the CA1 cell, it was inferred that plasticity had occurred at synapses from CA3 place cells, which had been active within a window of several seconds around the PP. Subsequent work using optogenetic stimulation of CA3 inputs to CA1 cells has since directly revealed the synaptic basis of this mechanism, termed Behavioral Time-Scale Plasticity (BTSP) [8]. Large-scale imaging studies suggest that BTSP plays a major role in the formation and consolidation of place cell activity at the network level [7–9]. BTSP is therefore a strong candidate mechanism for shaping network-wide patterns of connectivity in the hippocampus in one shot, as likely occurs in the formation of episodic memories.

A computational model of BTSP was developed which succeeded in reproducing the observed emergence of place fields from in-vivo experiments [6, 10]. In the model, plasticity occurred due to the temporal overlap of an eligibility trace at each synapse [11], proportional to the presynaptic firing rate, with a global, instructive signal related to the PP. Fits of the model to the data also revealed that the observed changes in pre-existing place fields due to additional PPs were consistent with a synaptic weight-dependent rule. Specifically, potentiation of synapses which were active within the plasticity window dominated when the weights were initially weak, whereas strong synapses underwent depression. The place field resulting from this BTSP rule could be calculated analytically if presynaptic place fields were assumed to be rectangular in shape [12], but a more general analysis of such a nonlinear, time-dependent process is not straightfoward. Recent work has also shown that a simplified BTSP rule in which synapses are taken as binary can allow for the storage and retrieval of large numbers of binary input patterns in a feedforward architecture reminiscent of CA1 [13].

In this manuscript we show that this model of BTSP can, in fact, be reduced to a one dimensional map for the synaptic weight matrix. Namely, the state of any given synapse in the weight matrix can be expressed as the same weight at a previous time, plus a change due to the plasticity process. The map describes the weight changes after a single "event", for example the occurrence of a PP during a single traversal of a track. The advantage of this formulation is that it allows for an extensive and detailed mathematical analysis of the effect of BTSP on the network connectivity. We first show that the map can qualitatively reproduce findings from the induction of PPs in CA1 pyramidal cells in mice. We then leverage the analytical tractability of the map to investigate how BTSP might shape the recurrent excitatory weights in a network such

as CA3 during the exploration of novel environments. Specifically, we consider a sequence of a large number of circular tracks and compute the correlation of the weight matrix with each one, as a function of its age, as well as the variability in this correlation due to the interference between different tracks. After the learning process, we use the full weight matrix to run simulations in a network of rate neurons, and quantify the memory capacity of the network based on the number of tracks for which bump attractors can be spontaneously generated. Finally, we show, using the analytically derived connectivity statistics, that the dynamics of the full network can be closely approximated by a set of ring models, one for each track, as long as the coding is sufficiently sparse. We conclude that BTSP is well suited as a mechanism for one-shot encoding of memories as attractors in recurrent networks.

## Results

Previously silent CA1 pyramidal cells can suddenly become place cells after the occurrence of a plateau potential (PP). Experimental evidence suggests that the PP effectively "switches on" synapses from spatially tuned CA3 inputs, leading to a tuned subthreshold membrane potential in the CA1 cell [6, 8]. This phenomenon is illustrated in Fig 1. As the animal runs along a linear track, some CA1 cells receive little or no CA3 input due to the ineffectiveness of the synaptic connections, see Fig 1A. The resulting membrane potential of the CA1 cell is therefore initially spatially untuned, Fig 1B. When a PP occurs in the CA1 cell at a given location along the track (see PP symbol in Fig 1A), synapses from CA3 place cells which are active within a window of a few seconds around the PP become potentiated, leading to spatial tuning, Fig 1C and 1D.

A computational model of this process has been successful in describing the changes in membrane potential observed in CA1 cells after the induction of a PP at a given location along a virtual track [10]. The model keeps track of both a synaptic elegibility trace, related to the activity of CA3 cells presynaptic to the CA1 cell of interest, as well as of an instructive signal, related to the occurrence of the PP. The instructive dendritic signal is global in nature, allowing for the possibility of all activated synapses to be updated simultaneously. The resulting plasticity at a given synapse depends on the convolution of these two signals, passed through a nonlinearity and integrated over the lap. There is a different elegibility trace for potentiation and depression. For details of this biophysical model, see Methods as well as [10]. Here we show that the total plasticity which occurs over a lap from this model can be described as a map in a straightforward way. Namely, the synaptic weight from a cell $j$ to a cell $i$ on a lap $k$ can be written

$$w_{ij}^k = w_{ij}^{k-1} + \Delta w_{ij}^k, \qquad (1)$$

where the change in the weight due to the occurrence of a PP on lap $k$ does not explicitly depend on continuous time, but rather only on the identity of the presynaptic and postsynaptic neurons. Biophysically, the degree of potentiation (or depression) depends on the convolution of the synaptic elegibility trace and the instructive signal, which is maximal when the PP and presynaptic activity happen at the same time. If the presynaptic activity occurs far before, or far after the PP, plasticity will be small or non-existent. Hence we can plot a continuous curve of the strength of plasticity as a function of the time difference between the activation of a presynaptic cell and the time of the PP, $\bar{f}(t)$ see Fig 1E (left) where it is assumed that the PP occurs at time $t = 0$. As the animal runs along the track and presynaptic place cells are activated in order, we can assign a time difference to each cell index $j$. If the animal runs at a constant velocity $v$, then we can directly convert time to space by multiplying by the speed to obtain $\bar{f}((x - x_{PP})/v)$, where $x_{PP}$ is the position of the PP, see Fig 1E (bottom). One confound when

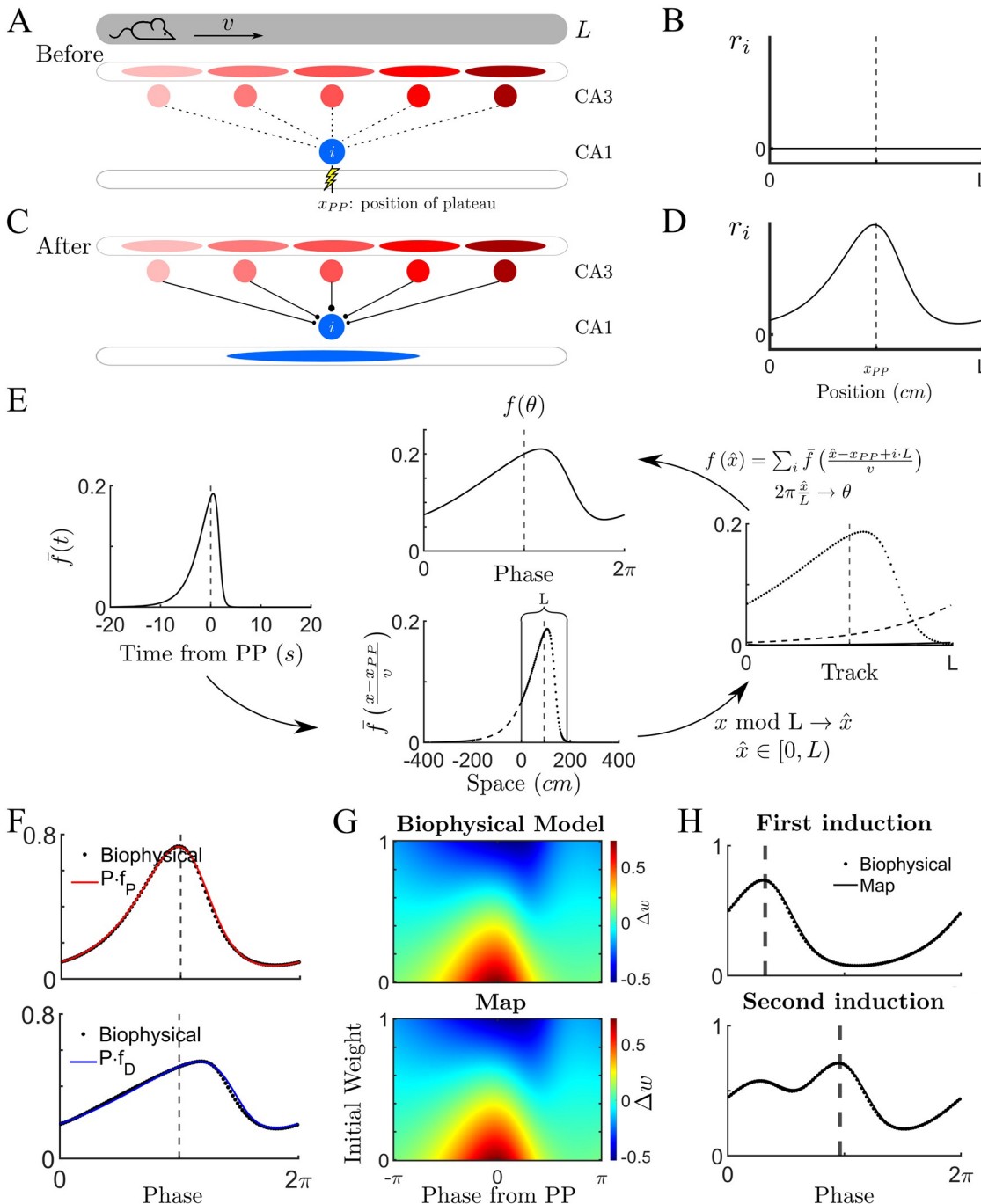

**Fig 1. A 1D map for BTSP can reproduce results from a biophysical model.** **(A)** A mouse runs at constant velocity $v$ on a virtual linear track of length $L$. Before the trial, $N$ CA3 place cells are weakly connected to the postsynaptic CA1 cell $i$, i.e. the synapses are ineffective. On a subsequent trial, a dendritic plateau potential (PP) occurs inside the CA1 cell when the animal reaches a position $x_{PP}$. **(B)** The firing rate of cell $i$ before the PP is zero over the length of the track. (The dashed line indicates the position of mouse when the PP occurs.) **(C-D)** After the occurrence of the PP, synapses from CA3 place cells are potentiated. As a result, the CA1 cell $i$ develops spatial tuning. **(E)** A temporal plasticity kernel from the biophysical model can be converted to a spatial kernel for a 1D map. Left: The biophysical model can be fit to a temporal plasticity kernel extracted from experiment. Bottom: For an animal running at a constant velocity, the temporal kernel can be expressed as a spatial kernel with a linear transformation. Right: For running on a virtual track with teleportation, the spatial kernel must be "wrapped". Top: A phase can be defined by normalizing the spatial kernel by the track length. **(F)** We define $f_P$ (red line) and $f_D$ (blue line) as the normalized spatial plasticity functions. Here they are compared to results from simulations of the biophysical model, also expressed as a function of phase along the track. **(G)** The degree of plasticity inferred from experiment depends not only on the plasticity functions, but also on the previous synaptic

strength, normalized here to lie between 0 and 1. (H) Once the map has been fit to the biophysical model for a single PP, we can test the robustness by comparing results for more complex protocols. Here we see the result after a second PP occurs. See S1 Fig for additional induction protocols. See Methods for parameter values.

converting from time to space is the fact that while time extends along the whole real axis, space is restricted to lying between 0 and the total track length $L$ (in the virtual reality set-up, the animal is teleported back to the beginning of the track). In practical terms this means that the plasticity of the synapse from a given presynaptic cell may have several contributions at different points in time. Specifically, portions of the spatial plasticity rule lying outside the track region (below 0 and greater than $L$) need to be wrapped around to fit within the domain, see Fig 1E (bottom and right). This results in a spatial plasticity function, which is the sum of the wrapped contributions of the temporal plasticity function, see Methods for details. Finally, we normalize the track length to define a phase and obtain the two spatial plasticity functions $f_P(\theta)$ and $f_D(\theta)$ for potentiation and depression respectively, see Fig 1E (top). To fit the simulations from the biophysical model in Fig 1, we used a wrapped skew-t distribution (solid lines in Fig 1F), see Methods.

Once the spatially-dependent plasticity functions $f_P$ and $f_D$ are determined, the synaptic weight change is given by

$$\Delta w_{ij}^k = P(1 - w_{ij}^{k-1}(\theta_j))f_p(\theta_j, \bar{\theta}) - Dw_{ij}^{k-1}(\theta_j)f_D(\theta_j, \bar{\theta}), \tag{2}$$

where $\theta_j$ is the centroid of the place field of the presynaptic CA3 cell $j$, $\bar{\theta}$ is the phase at which the PP occurs, and the parameters $P$ and $D$ control the strength of potentiation and depression, respectively. See Methods for a detailed description. The synaptic weight change from the map, plotted as a function of the position of the presynaptic cell and the value of the synaptic weight before plasticity, closely matches that of the biophysical model, Fig 1G. The spatially-dependent plasticity functions fit for a single PP can be used to determine the synaptic weights following a series of inductions, which closely match those found through numerical simulation of the biophysical model, Fig 1H. Also note that the same temporal plasticity function will give rise to distinct spatial functions for different running velocities: narrow functions for low velocity and broader ones for larger velocities, see S1 Fig for a detailed description of this effect as well as additional, more complex plasticity protocols and other sets of parameters. The fact that running speed modulates the width of emergent place fields is observed experimentally, and is a hallmark of BTSP [6]. Note that if the running speed of the animal is not constant, then the mapping of the plasticity rule from time to space still exists, but will be nonlinear.

## BTSP and memory storage in recurrent networks

We can use the 1D map, Eq 1, to study the plasticity in a recurrent network of place cells, such as in area CA3 of the hippocampus. Specifically, we are interested in how the plasticity rule leads to the formation of stable internal representations of different spatial environments. Doing so requires two distinct steps. First, we must determine how plasticity shapes the matrix of recurrent connections. This we will do through direct analysis of the 1D map. In doing so, we will calculate the average correlation of the synaptic weight matrix with any given environment, as well as the degree of quenched variability. This will allow us to perform a signal-to-noise ratio calculation, and determine how the memory capacity scales qualitatively with network size, coding sparseness, and the learning rates $P$ and $D$ [14, 15]. Secondly, and importantly, we need to study how the connectivity shapes the activity in a network model, and determine the true memory capacity of the network in terms of stable attractor states [16, 17].

**1D map for BTSP in recurrent networks.** We first consider the simplest case in which all of the cells of the network have place fields in any given environment. We will study the more realistic case of sparse coding in a subsequent section. We assume that when the animal first explores a novel environment, place cells in CA3 are already present, or quickly form due to plasticity in afferent inputs. We can therefore order the cells along a linear track according to their place field location, even before any plasticity occurs in the recurrent connectivity, Fig 2A (before). In fact, the recurrent connections between place cells have been shaped by previous learning and are initially uncorrelated with the novel environment, Fig 2B (left). As the animal runs along the track, we assume that, over some number of traversals, PPs occur in all of the cells, and that the time of their occurrence coincides with the maximal firing rate of the post-synaptic cell. In the more realistic sparse-coding case studied in the next section, PPs will only occur in a small fraction of cells, which will determine the sparseness. In this way, strong potentiation of recurrent excitatory inputs is expected between cells with adjacent place fields, while cells with distant place fields may undergo no potentiation, or even depression, depending on the details of the plasticity functions and the previous state of the synapse in question. Therefore, the recurrent connectivity becomes correlated with the ordering of the cells' place field in the novel environment, Fig 2A (after) and B (right). Specifically, the update to the weight matrix in Eq 1 now takes the form

$$\Delta w_{ij}(\Delta \theta_{ij}^k) = P(1 - w_{ij}^{k-1})f_P(\Delta \theta_{ij}^k) - D w_{ij}^{k-1} f_D(\Delta \theta_{ij}^k), \tag{3}$$

where $\Delta \theta_{ij}^k$ is the phase difference in the place field positions of cells $i$ and $j$ in environment $k$.

We model the plasticity due to the exploration of a number $n$ of different linear tracks, Fig 2C. The ordering of the place cells along the track is randomly reshuffled from one environment to the next. Although synaptic weights are updated for any new environment explored, after a certain number of environments, which depends on the learning rates $P$ and $D$, the statistics of the weight matrix reach a steady state, see Fig 2D. The steady-state statistics can be calculated directly from the plasticity rule, e.g. the mean $\mu = \langle w_{ij} \rangle$ and variance, $\sigma^2 = \langle w_{ij}^2 \rangle - \mu^2$, where the brackets indicate averages of the network, see Methods for the detailed calculations.

At this point the learning process itself is stationary, and hence the correlation structure of the weight matrix with any previous environment depends only on how far in the past it was explored, and not its exact ID. As a measure of the correlation of the weight matrix with a given environment, we extract the first Fourier coefficient for the corresponding ordering of phases. That is, the average correlation with the environment $n - \eta$, where $n$ is the last environment explored, is given by

$$a_\eta = 2 \langle \cos(\Delta \theta_{ij}^{n-\eta}) w_{ij}^n(\Delta \theta_{ij}^{n-\eta}) \rangle, \tag{4}$$

and the mean synaptic weight of cells ordered in environment $n - \eta$ is $M_\eta = \mu + a_\eta \cos \Delta \theta_{ij}^{n-\eta}$. The parameter $\eta$ indicates how far back in the past the environment was explored. Note that in order to perform the averaging correctly in Eq 4 it is necessary to order the weights according to their phases in environment $n - \eta$. However, the average correlation alone is not sufficient to determine if information about a given environment can be reliably recovered. In fact, the learning process itself is stochastic due to the random global remapping of place fields from one environment to the next and the connectivity is therefore noisy. We quantify the strength

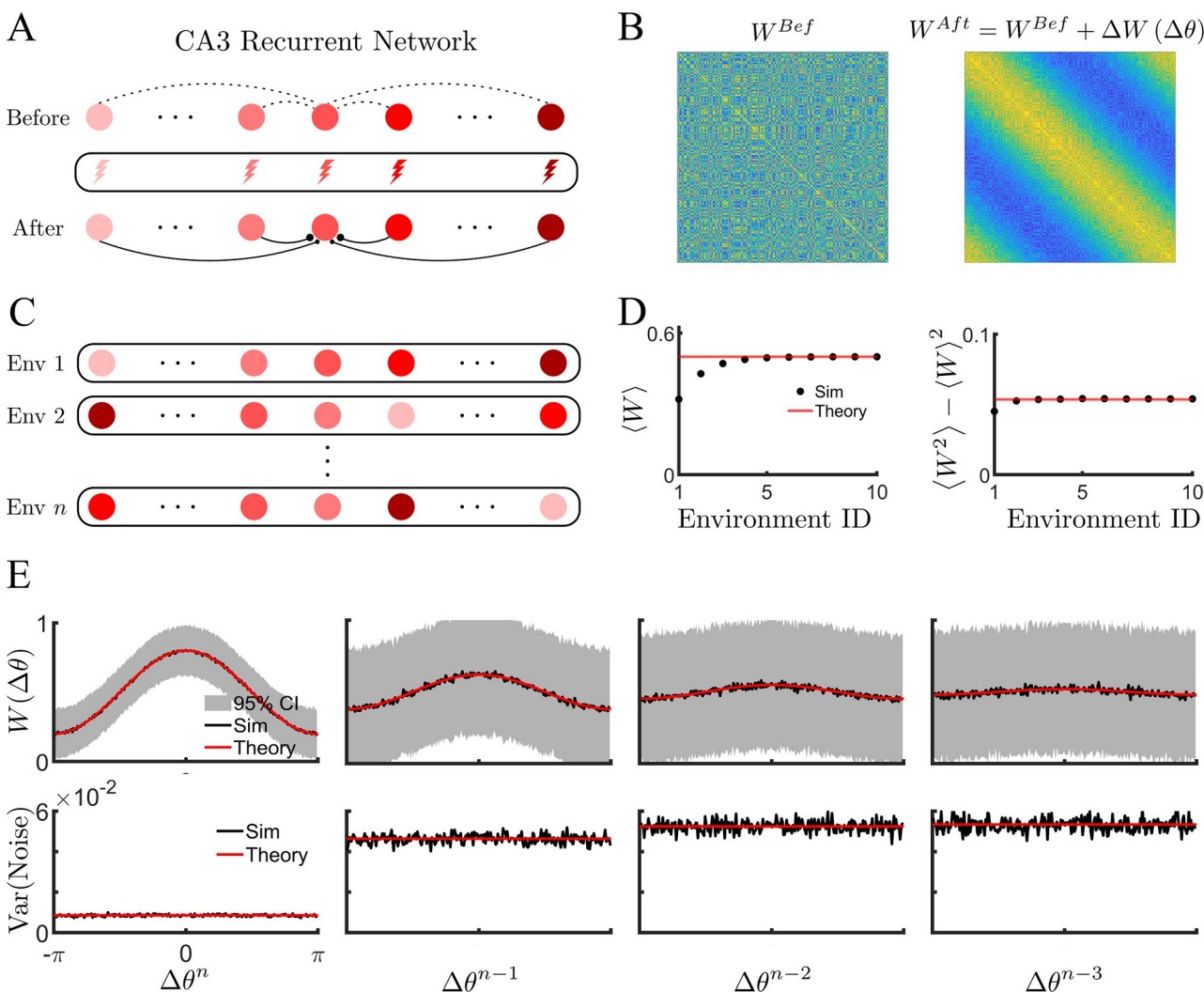

**Fig 2. BTSP correlates the connectivity in a recurrent network with past explored environments.** **(A)** Schematic of BTSP-based learning in a recurrent network. Before the learning: the recurrent connectivity in a population of $N$ CA3 place cells is initially uncorrelated with the novel environment. During exploration of the novel track: a dendritic plateau potential occurs inside each CA3 cell, driving plasticity. After: the synaptic weights are modulated according to the phase difference. **(B)** Ordering cells w.r.t. the novel environment before (left) and after (right) the BTSP-based learning reveals the emergence of correlation. **(C)** Schematic of plasticity for $n$ distinct environments. Global remapping occurs for each novel environment (random permutation of place field location). **(D)** Global statistics of the connectivity matrix, average (left) and variance (right), as function of number of explored environments. **(E)** Spatially ordered statistics of weight matrix (sorted and centered w.r.t previous explored environments) in the steady state. Top: Mean (black line) and 95% confidence intervals (grey lines) of simulated weight as function of phase difference at each environment, as well as the theoretical prediction (red line). Bottom: Variance of the ordered weights: simulation (black) and theoretical curve (red). **Parameters**: $N = 256$, $n = 50$, and $P = D = 0.3$.

of quenched variability by calculating the variance about the mean correlation. Namely,

$$V_\eta = \langle (w_{ij}^n - M_\eta)^2 \rangle. \tag{5}$$

In order to store memories as stationary attractors, we consider symmetric plasticity functions, which imply a temporally symmetric BTSP rule. An asymmetric rule could lead to the formation of dynamic attractors [18, 19]. For the choice of plasticity functions $f_P(\theta) = 1 + \cos\theta$ and $f_D(\theta) = 1 - \cos\theta$, the mean correlation takes on the simple form $a_\eta = \frac{2PD}{P+D}(1 - P - D)^\eta$,

while the variance can be expressed as $V_\eta = A_\eta + B_\eta \cos \Delta\theta + C_\eta \cos^2 \Delta\theta$, see Methods. Fig 2E shows a comparison of the theoretical results with numerical simulation of the plasticity rule for the last four environments explored: $\eta = 0, 1, 2, 3$. Note that the mean correlation decays as a function of the "age" of the memory, i.e. the bump in connectivity flattens. This is due to the overwriting of synapses from subsequent learning, the so-called "palimpsest" property, which is inevitable in biological learning systems [16]. Additional examples for other parameter values are shown in S2 Fig. On the other hand, the quenched variability in the network, visualized as the 95% confidence intervals of the ordered weights (top row, grey lines), does not decay away, although it does generally vary with the age of the memory, see bottom row.

**Sparse coding with BTSP in recurrent networks.** In the previous section we considered a network in which every neuron has a place field in every environment. Furthermore, each cell encoded a distinct phase in any given environment. The resulting network had a low storage capacity because there is both maximal interference between plasticity events from different environments, as well as a low signal-to-noise ratio (SNR) for any given location in an environment. Interference between memories can be reduced by considering sparse coding, for which only a fraction $s$ of neurons are place cells in any given environment. The SNR at a given location can be increased by consider a population of $M$ cells for each such location. Specifically, we consider a network of $NM$ cells, where $N$ is the number of uniformly distributed spatial phases, and M is the number of cells available to encode each of these phases. In the previous section we had $M = 1$.

We model BTSP in the recurrent connections as before, with the difference that, given a sparseness $s$, there are now $sM$ neurons encoding each spatial phase, Fig 3A. It is important to note that neurons which have overlapping place fields (belong to the same group of $sM$ neurons) in one environment, generally do not have overlapping fields in another. Namely, it is the individual cells which undergo global remapping, not the cell populations. Fig 3B illustrates how the sparseness $s$ and population size $M$ affect the resulting connectivity matrix after plasticity in a novel environment. Increasing $M$ results in a block structure, Fig 3B (upper row), which leads to an increased SNR. Specifically, if we consider that each post-synaptic neuron receives the average synaptic weight of a pre-synaptic population (block), then the amplitude of the correlation remains independent of the population size, while the variance decreases like one over the population size, Fig 3C (left). The SNR, and hence the memory capacity, both increase with increasing $M$ for fixed $s$, Fig 3D (left). On the other hand, for $s < 1$, some fraction of the synaptic weights do not undergo plasticity in a novel environment, Fig 3B (bottom). As a consequence, there is less interference between memories, and the decay of the correlation amplitude becomes shallower for lower $s$, Fig 3C (right). The effect on the variance is non-trivial, but is captured by the theory, Fig 3C (bottom right). The shallower decay in the amplitude for sparse networks leads to a larger SNR at long times compared to a network with the same population size, but larger $s$, Fig 3D (upper right). As a result, the memory capacity increases dramatically in the sparse-network limit, Fig 3D (lower right).

The SNR can be used to calculate the memory capacity analytically. Specifically, we define the capacity $\eta_{cr}$ where $SNR = \frac{a_{\eta_{cr}}}{\sqrt{V_{\eta_{cr}}}} = T$, where the threshold $T$ is of order one. When potentiation and depression are balanced ($P = D$) and assuming that the maximum capacity $\eta_{cr} \gg 1$, the formula for the SNR simplifies significantly and we find that

$$\eta_{cr} \sim \frac{1}{4Ps^2} \ln (sM), \tag{6}$$

see Methods. The scaling with the logarithm of the system size is a classical result for networks with bounded synapses [14, 16], as is the dramatic improvement in capacity through sparse

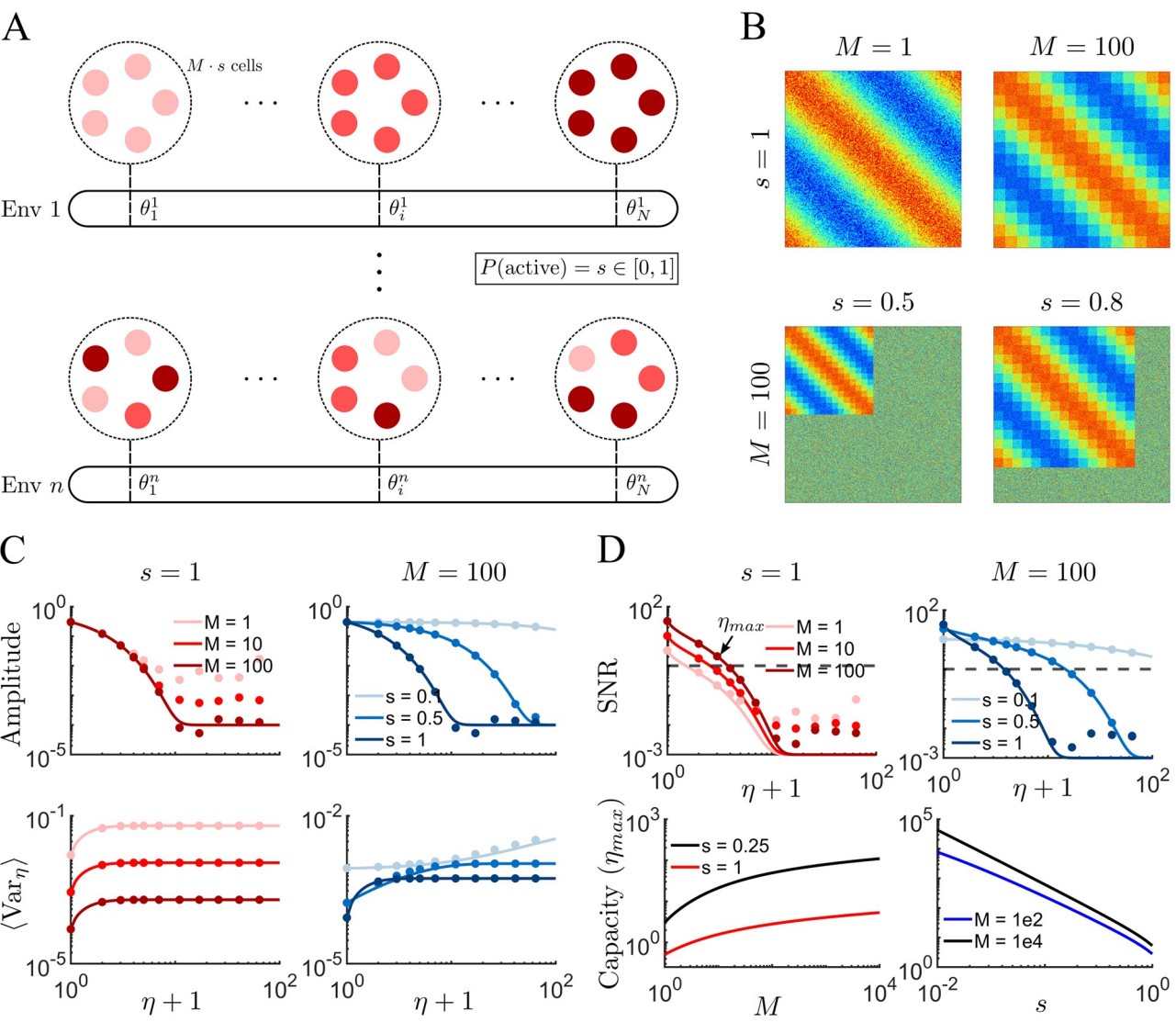

**Fig 3. Memory capacity improves with sparse coding and increased population size.** (A) Schematic of the recurrent network with population size $M$, and sparseness $s$. For each cell, the probability of being a place cell in any given environment is $s$. There is a population of $sM$ cells with place field centered at position $\theta_i$. For each environment, there is a global remapping (reshuffling) of cell positions. (B) The weight matrix for different values of $M$ and $s$. (C) Amplitude of the mean correlation $a_\eta$ (Fig 2E top) and of the variance $V_\eta$ (Fig 2E bottom) as a function of number of past explored environments in log-log scale. (D) Top: SNR for cases studied in panel C. Dashed line indicates $SNR = 1$. Bottom: The memory capacity of system, $\eta_{max}$, as function of population size (left) and sparsity (right). **Parameters**: $N = 128$, $n = 125, 200, 1000$ for $s = 1, 0.5, 0.1$ respectively, and $P = D = 0.3$. To see the effect of varying $P$ and $D$ on the SNR, see S3 Fig.

coding [16, 17, 20, 21]. Simply put, if additional synaptic resources are available to increase the SNR, the best strategy would be to fix $sM$ while decreasing $s$, i.e. make the representation sparser. This means that $s \sim 1/M$ and hence the memory capacity $\eta_{cr} \sim M^2$, a scaling shared by a number of memory systems with sparse coding [22]. Additional strategies for improving capacity, including allowing for multiple, interacting discrete synaptic states, have been studied in detail elsewhere [15, 23–25].

Note that the SNR scales with the population size at each position along the track, $sM$, and not with the number of positions $N$. In fact, our choice of continuous spatial plasticity functions $f_P(\theta)$ and $f_D(\theta)$ implicitly assumes that $N$ is large enough for this approximation to hold.

In theory this sets a lower bound on $N$ dictated by the sampling theorem. Beyond this constraint, $N$ does not affect the SNR because the SNR is set by the degree of redundancy in the input from cells with the same, or very similar tuning, which is $sM$. However, we will find that once we consider the recall of attractor states in neuronal networks, the number of encoded positions $N$ will play an important role.

## Neuronal networks endowed with BTSP can encode a large number of spatial memories as bump attractors

We have studied the statistical properties of the recurrent synaptic weights in networks with BTSP. We have calculated how the number of spatial environments which can be encoded scales with the system size and sparseness. In doing this, we have implicitly assumed that a subset of cells is active in any given environment, but we have not explicitly modelled the firing rate dynamics. Rather, we have directly solved the 1D map. Nonetheless, the true memory capacity is determined by the recovery of intrinsic dynamical states which are correlated with the neuronal activity in any given environment. To this end we simulate networks of firing rate neurons with connectivity matrices determined directly from the 1D map for BTSP from the preceding section. Specifically, we consider now the firing rate dynamics in a network in which plasticity has already occurred, and no longer allow for additional plasticity.

In the network simulations, for a neuron $i$, the firing rate obeys

$$\tau \dot{r}_i = -r_i + \phi\left(\frac{1}{\kappa N}\sum_{j=1}^{MN} \bar{w}_{ij} r_j S_j^k + I_0\right) S_i^k, \qquad (7)$$

where $\kappa$ scales with the local population size $sM$ and $\phi$ is a nonlinear fI curve, see Methods for details. The parameter $S_i^k = 1$ if neuron $i$ is a place cell in environment $k$ and is zero otherwise. The synaptic weight $\bar{w}_{ij} = W_0 + W_{max}(w_{ij} - \mu_w)$, where $w_{ij}$ is taken directly from the 1D map after the learning process, $W_{max}$ is the maximal synaptic weight ($w_{ij}$ is normalized to be bounded between 0 and 1), and $\mu_w = \langle w_{ij}\rangle$. We model the effect of inhibitory interneurons by assuming a global inhibitory feedback proportional to the mean excitatory activity. This effect is included in the mean offset $\mu_w$ and in the parameter $W_0$, see Methods for details of the network and parameter values. We find that such a network can indeed encode a large number of attracting bump states. Fig 4A shows three illustrative simulations from the same network with three different initial conditions. Each initial condition consists of a constant input to the subpopulation of neurons which were active in that environment during the learning process, indicated at the top of the corresponding panel, as well as a small, bump-shaped perturbation. The dynamics evolve to a steady state and the activity is then visualized by ordering the neurons according to their phases in either the last environment explored $n$ (top), second to last $n - 1$ (middle) or ten environments ago $n - 10$ (bottom). Depending on the initial condition, a coherent bump can be recovered in the corresponding space, while the same activity appears as noise given another ordering.

These simulations suggest that the dynamics in the network of firing rate neurons with the full recurrent connectivity might be approximated by considering a set of distinct ring models, one for each past environment, Fig 4B. The network connectivity itself has highly complex structure, but when projected onto the subspaces corresponding to different environments it is largely characterized by the mean correlation with that space, e.g. the amplitude of the first Fourier coefficient, and the degree of quenched variability due to interference between environments. These are precisely the quantities that we calculated analytically in the previous section. Specifically, in the ring model for an environment $n - \eta$, we can write the connectivity

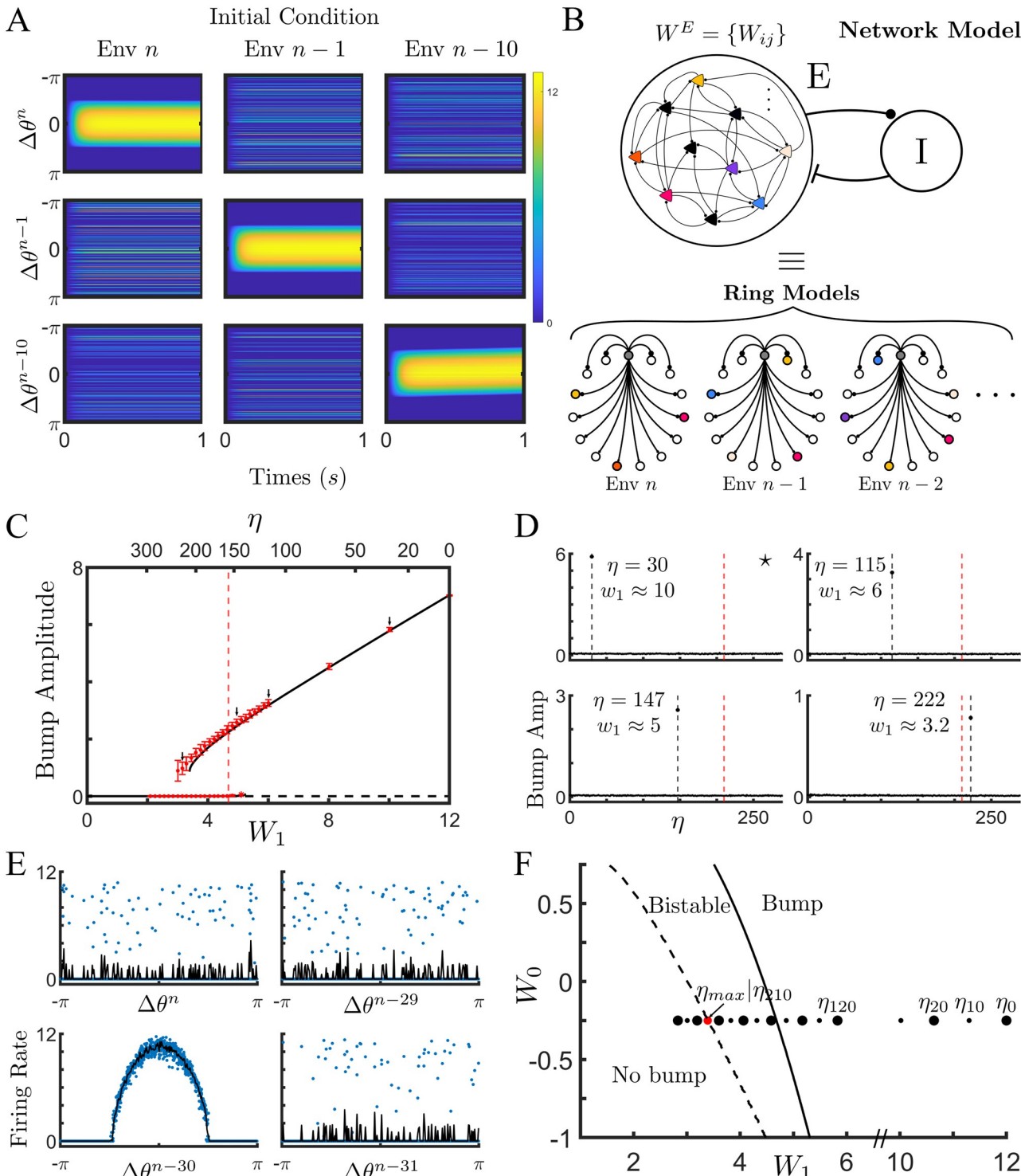

**Fig 4. Network dynamics shaped by BTSP in the presence of sparse coding are equivalent to a hierarchy of weakly-interacting ring models. (A)** Illustration of the bump dynamics for a single network with three distinct intial conditions. **(B)** The topology of the recurrent network shaped by BTSP, and with sparse coding, can be approximated by a series of rings, one for each environment. **(C)** Bifurcation diagram as function of $W_1$ (bottom x-axis) or $\eta$ (top x-axis) for the ring model and the network model respectively. Lines indicate the stable (solid line) or unstable (dashed line) solution of the ring model. Red dots are the mean amplitude of bump solutions from the network model, and error bars give the 95% confidence interval. Dotted line represents the analytically calculated critical value of $W_1$ at the Turing bifurcation for the ring model. **(D)** Bump amplitudes calculated by ordering cells according to their preferred positions for a range of environments. The amplitudes are averages over ten simulations with distinct matrices. The four

panels correspond to values of $\eta$ indicated by arrows in C. Black vertical dashed line: retrieved environment. Red dashed line: oldest retrievable environment according to ring-model approximation. **(E)** The steady-state neuronal activity in the full network ordered in four different environments for the retrieval of environment $n - 30$. This is the case marked with an asterisk in D. **(F)** Phase diagram. Solid line: Turing bifurcation. Dashed line: saddle-node of bump solution. Dots show values of $W_1$ for different $\eta$s of the network model when $W_0 = -0.25$. **Parameters**: $N = 256$, $n = 1500$, $M = 60$, $s = 0.1$, $P = D = 0.3$, $W_{max} = 40$, and $W_0 = -0.25$.

between neurons at a phase difference of $\Delta\theta^{n-\eta}$ as

$$W(\Delta\theta^{n-\eta}) = W_0 + W_1^{\eta}\cos(\Delta\theta^{n-\eta}) + \Delta W z(\Delta\theta^{n-\eta}), \tag{8}$$

where $W_1^{\eta} = W_{\max}\frac{sM}{\kappa}a_{\eta}$, $\Delta W = w_{\max}\frac{\sqrt{sM}}{\kappa}\sqrt{V_{\eta}}$, and $z$ a zero-mean Gaussian random variable with unit variance. The advantage of the ring-model formulation is that it is low-dimensional, and hence much simpler to analyze than the full network Eq 7. For example, if we choose the scaling $\kappa = \sqrt{SM}$ in the network (and with $P = D$) it can be shown, through analysis of the ring model, that the memory capacity, as estimated through a linear stability analysis of the spatially uniform steady-state, scales exactly as in Eq 6, see Methods. However, in numerical simulations we have taken $\kappa = sM$.

Eq 8 is therefore a mapping between the statistics of the connectivity in the full network, and the coupling parameters in the ring model. Specifically, in the full network we choose an ordering of the neurons corresponding to environment $n - \eta$, i.e. age $\eta$. We then run simulations and measure the bump amplitude after the steady state is reached. Because the connectivity matrix has quenched variability due to the learning process, we generate ten different matrices using distinct random orderings for the remapping of place fields. For each of the ten matrices we run simulations for the same value of $\eta$ with two different initial conditions: a small-amplitude and a large-amplitude bump. We then plot the mean and 95% confidence interval of the final bump amplitude in Fig 4C (symbols). For each value of $\eta$ we can calculate the corresponding connectivity in the ring-model using Eq 8. Changing $\eta$ has two effects: the first is that the spatial modulation in the recurrent connectivity $W_1$ decreases with increasing age, and the second is that the strength, and shape of the quenched variability also changes, see Eq 32. Fig 4C shows a comparison of the bifurcation diagram generated using the full network model (symbols) with the low-dimensional approximation given by the ring model (lines). The agreement is good for this value of sparseness $s = 0.1$, although there are discrepancies near the bifurcation points. We calculated the critical value of $W_1$ at which a Turing instability, from a flat state to a bump attractor, would take place (vertical dotted line). This calculation takes the quenched variability of the connectivity into account, see Methods for details. The upshot is that static variability in the connectivity effectively shifts the bifurcation to lower values of $W_1$. Note here also that the initial Turing bifurcation is supercritical, i.e. continuous, leading to very small amplitude but stable bumps just to the right of the predicted bifurcation, see S4(A) Fig. In fact, it can be shown through a weakly nonlinear analysis that for a value of $W_0 < 0$ this will always be the case. However, the small amplitude bump becomes unstable in a secondary bifurcation, leading to a large amplitude bump solution and a region of hysteresis in Fig 4C. Interestingly, the saddle-node bifurcation of bump solutions is shifted even further left in the full network model compared to the ring model. It is this bifurcation which determines the true memory capacity of the network, since below this point no bump solutions are possible. This discrepancy is likely due to the additional source of variability in the network model: the distribution of firing rates within each population. In the ring model, only the variability in the connectivity is accounted for, while the firing rate at each position represents the mean activity over the population of cells with that preferred location.

It is important to emphasize that the x-axis in the bifurcation diagram in Fig 4C refers to the age of the memory $\eta$ and hence each point corresponds to a different low-dimensional manifold. In the case of the ring model (lines) the solution will always be restricted to this manifold by definition, but that need not be the case for the full network model. Put another way, the y-axis measures the amplitude of a bump with the ordering corresponding to a particular $\eta$, but other orderings are possible, and hence bump amplitudes can be measured for all explored environments. To make sure that the bump solutions shown in Fig 4C are bumps only in the desired environment $\eta$, we measured the bump amplitude over a range of $\eta$s for several example cases. Fig 4D shows that for the four chosen values of $\eta$, indicated as well by arrows in Fig 4C, the bump exists only on a single manifold, while projections of the activity onto other manifolds yield disordered activity (bump amplitude zero). Fig 4E shows an example the steady-state bump solution when a small amplitude bump is seeded according to the ordering of neurons for $\eta = 30$. As can be seen, the bump remains constrained to the manifold for $\eta = 30$. Finally, the capacity of the network can be read off from a phase diagram of the dynamical states, Fig 4F. Specifically, the dashed line indicates the saddle-node line of bump states, to the left of which no bump states exist. The solid line is the Turing bifurcation itself, while between these two lines any bump state which exists is also bistable with the flat state, in as far as the ring-model approximation is accurate. The effect of varying the quenched variability $\Delta W$ and the external drive $I_0$ on memory capacity are shown in additional phase diagrams, see S4(B)–S4(D) Fig.

**The role of system size on quenched variability in the ring model.** We have seen that the quenched variability, calculated from the plasticity rule, takes the form $\Delta W(\theta) = \frac{\sqrt{sM}}{\kappa} \sqrt{V(\theta)} z(\theta)$, where $\theta$ is the phase difference between two neurons in the environment of interest, and $z$ is a spatially uncorrelated random variable drawn from a Gaussian distribution with zero mean and unit variance. In general, then, the strength of the variability is determined by the population size at each position $sM$. For example, if we choose $\kappa = sM$, then the quenched variability will vanish with increasing population size. The number of encoded positions $N$ does not play a role here.

However, this situation changes dramatically in the vicinity of an instability of a spatially modulated mode, e.g. at a Turing bifurcation. To see this, we expand the variability as a Fourier series on the ring. Setting aside the prefactor of $\frac{\sqrt{sM}}{\kappa}$, we have

$$\sqrt{V(\theta)} z(\theta) = 2\sum_{j=1}^{N} \alpha_j \cos(j\theta) + 2\sum_{j=1}^{N} \beta_j \sin(j\theta), \tag{9}$$

where $N$ is the number of encoded positions. The coefficients $\alpha$ and $\beta$ are Gaussian random variables with mean zero, and variances, and co-variances which must be determined self-consistently. In order to do this we need to make use of Parseval's theorem, which states that the variance, or power, in a signal must be conserved in Fourier space. We will also use the fact that $z$ is white-noise process in space, which implies that the power is not only conserved, but that it is distributed evenly across all modes. Near the Turing bifurcation there is an instability of the first Fourier mode ($j = 1$), which therefore is the mode we should be concerned with. We find that the variances of $\alpha_1$ and $\beta_1$ are $(A + 3C/4)/(2N)$ and $(A + C/4)/(2N)$ respectively, recalling that $V(\theta) = A + B\cos(\theta) + C\cos^2(\theta)$, see Methods and S2 Appendix for a detailed explanation. There is a pair of Fourier coefficients $\alpha_1$ and $\beta_1$ for each place cell in the network, which contributes a term $R_1 \cos(\theta - \phi_1)$ to the connectivity, where $R_1 = \sqrt{\alpha_1^2 + \beta_1^2}$, and $\phi_1 = \arctan(\beta_1/\alpha_1)$. These amplitudes and phases are, of course, also random variables, distinct for each neuron in the network. Therefore, at some locations in the network the quenched

variability will facilitate the instability to a bump solution (wherever $\phi \sim 0$), while at others this instability will be hindered ($\phi \sim \pi$). The net effect is to shift the instability to lower values of $W_1$, see Methods and S2 Appendix.

In any case, we can see that the shift in the bifurcation is inversely proportional to the system size $N$. Namely, smaller values of $N$ result in a larger memory capacity. On the other hand, if $N$ is too small, then the continuum approximation breaks down and no Turing instability would be expected to occur. This implies that there is an optimal encoding of position $N$ which maximizes memory capacity, all other parameters being the same. Fig 5 shows this effect for simulations of the network model using the same parameter values as in Fig 4.

**Mixed attractors for non-sparse coding.** The agreement between network simulations and the ring model approximation holds in the limit of sparse coding. In this limit, the overlap between sub-populations of place cells which encode distinct environments, vanishes. Hence the dynamics for the memory of each environment evolves on distinct, weakly-interacting manifolds. The weak interaction takes the form of quenched variability. For $s = 0.1$ (and $s = 0.2$, see S5 Fig) this approximation holds for all values of $\eta$, Fig 4C–4F. However, increasing the value of $s$ leads to nontrivial effects which are not captured by the ring model, including the emergence of new, mixed attractors. This phenomenon is illustrated in Fig 6. Bifurcation diagrams for $s = 0.3$ and $s = 0.5$ are shown in panels A and D respectively, where the lines are from simulation of the ring model, and the red circles are from simulation of the full network. As long as the memories are recent, the attractor states are restricted to a manifold corresponding to a single environment, see Fig 6B for $\eta = 3, 12$ and $16$ and E for $\eta = 1$. This is no longer the case for more remote memories, which generically form mixed states when $s$ is large enough, see Fig 6B for $\eta = 26$ and E for $\eta = 4 - 6$. The mixed states are states with spatial modulation in several, oftentimes many environments at the same time. For example, for $s = 0.3$ and $\eta = 26$ we see that the steady-state solution has significant spatial modulation in environment $\eta$, but also in at least the ten most recently explored environments, see Fig 6B and 6C. In

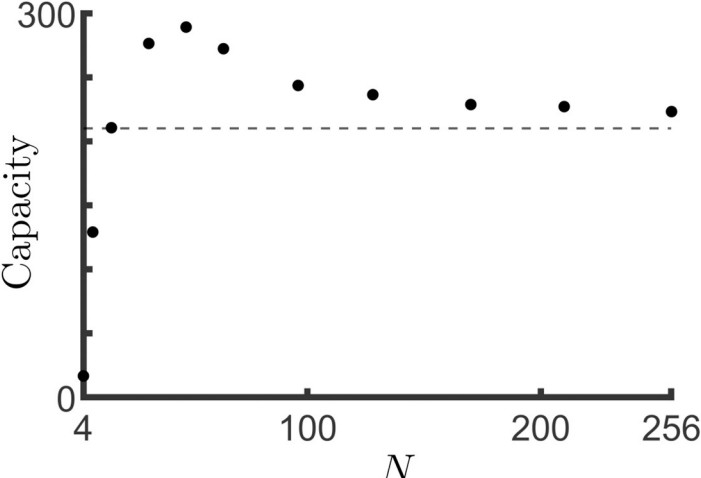

**Fig 5. The memory capacity of the network depends non-monotonically on the resolution of spatial tiling of place cells.** The memory capacity, calculated as the number of retrievable bump solutions in the network model, as a function of the number of encoded positions, i.e. the spatial resolution. In the large $N$ limit the capacity approaches that of the ring model without quenched variability, shown by the dashed line. As $N$ decreases, the amplitude of the quenched variability increases, facilitating the generation of bump solutions. Finally, when $N$ is too small, bump solutions no longer exist as the ring model approximation is not valid. The result is an optimal value of $N$ for which the memory capacity is maximal. Parameters are the same as in Fig 4.

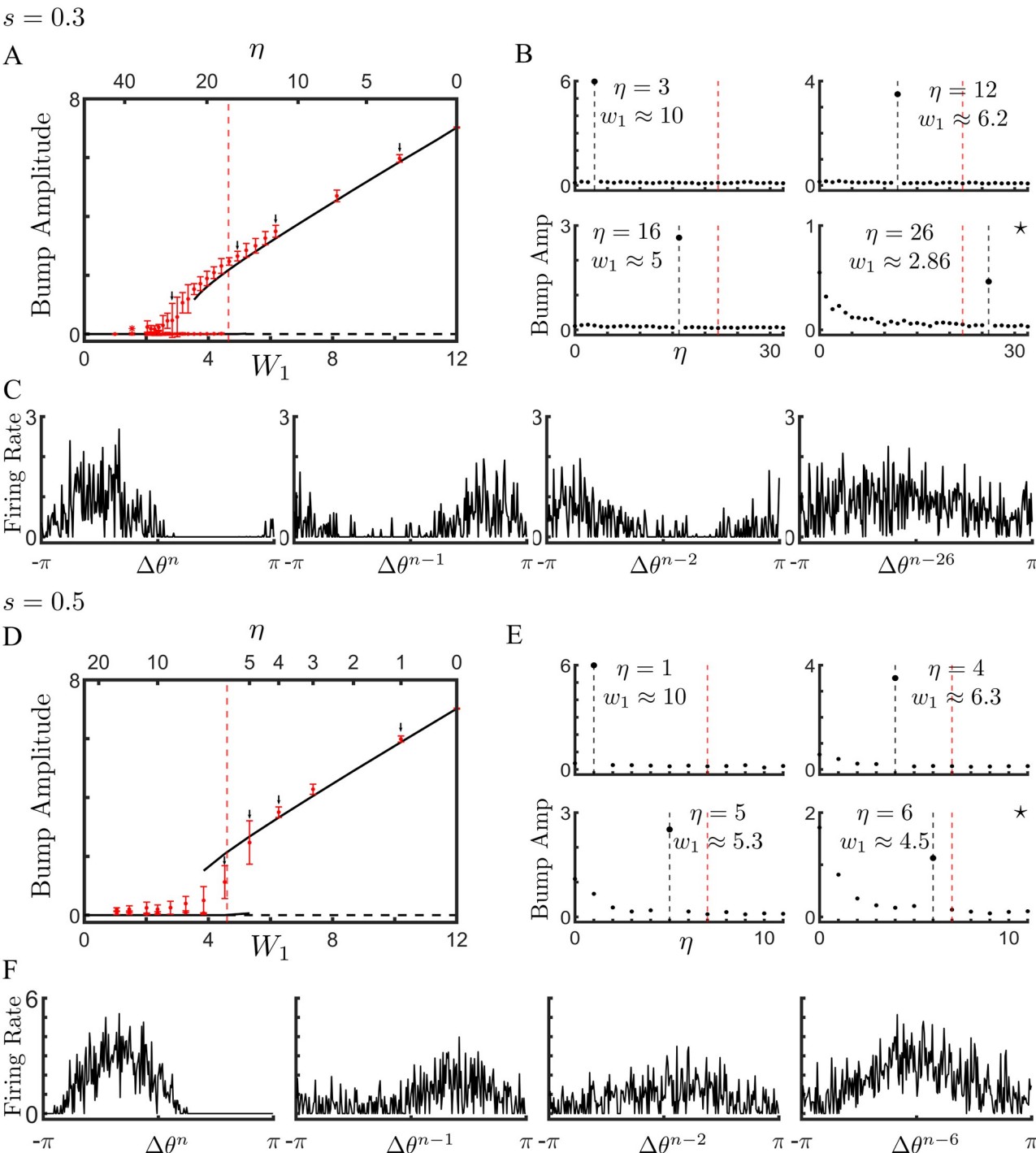

**Fig 6. Deviations of the network model from the ring-model approximation are due to the emergence of mixed attractors.** **(A)** Bifurcation diagram for $s = 0.3$. Lines are from simulation of the ring model, while symbols indicate the mean and 95% confidence intervals from simulations of the network model. **(B)** Amplitude of the bump as measured with different orderings. Note that for $\eta = 26$ the steady-state solution is a bump in many different environments at the same time. **(C)** Sample profiles of the same steady-state solution, but with distinct neuronal orderings corresponding to different environments. **(D-F)** Same as A-C but with $s = 0.5$. **Parameters**: $N = 256$, $n = 500$, $P = D = 0.3$, $sM = 6$, $W_{max} = 40$, and $W_0 = -0.25$.

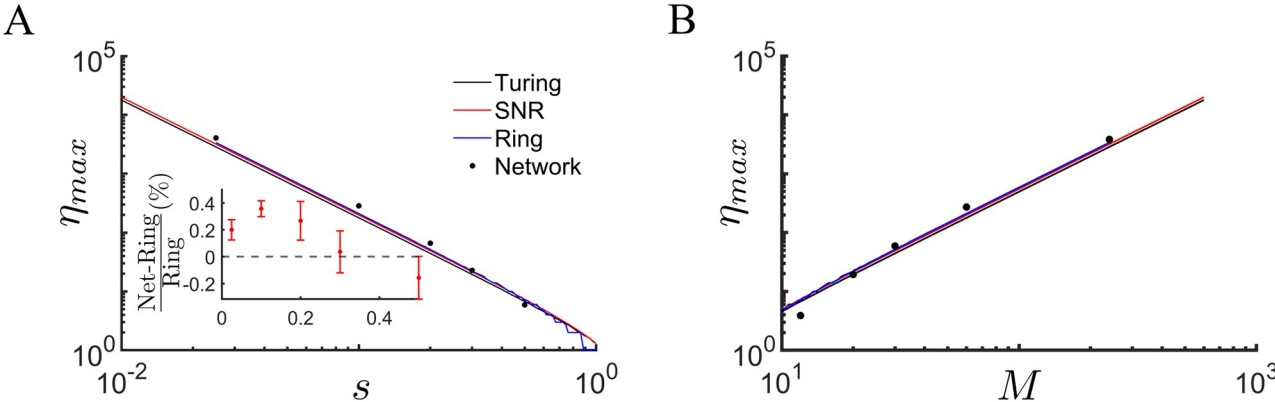

**Fig 7. Memory capacity in the network model outperforms ring model approximation and scales optimally with system size. (A)** The memory capacity $\eta_{max}$ as a function of the coding sparseness $s$. Black line: Instability to Turing Bifurcation, Eq 39. Red line: SNR calculation, Eq 34. Blue line: Numerical estimation of the saddle-node of bump attractors from the ring model. Dots: Capacity of full network. Inset: The fractional difference between the memory capacity of the full network and that of the ring, $\Delta\eta_{max} = (\eta_{max}^{network} - \eta_{max}^{ring})/\eta_{max}^{ring}$. The network outperforms the ring model in the sparse-coding limit. **(B)** The memory capacity as a function of the number of place cells encoding each position, $M$.

general, ithe regime of mixed attractors, the degree of variability between simulations using distinct realizations of the connectivity matrix is large, see the error bars in Fig 6A and 6D.

We conclude that a recurrent network endowed with BTSP can encode a large number of attractors through one-shot learning, as long as the coding is sufficiently sparse. In this limit, the interaction between attractors takes the form of quenched variability. Surprisingly this variability actually *enhances* the memory capacity by stabilizing bump attractors in parameter regimes for which they do not exist in the zero-noise case. The memory capacity in this sparse coding regime scales as $M^2$, where $M$ is the available number of place cells for encoding each position (only a fraction $s$ are active), see Fig 7, a theoretical optimum [22]. Finally, the full network endowed with BTSP actually outperforms the ring-model approximation in this regime, likely due to the additional quenched variability in the firing rates within each population of place cells, see inset Fig 7.

## Discussion

We have shown that a biophysical model of BTSP [10], developed to reproduce the results of in-vivo plasticity protocols in area CA1 of awake behaving mice [6], can be reduced to a 1D map. The map, which describes the updating to the synaptic weight matrix after a plateau-potential-driven plasticity event, can be fit to the biophysical model quantitatively. The advantage of the map is that it is a powerful mathematical tool which allows us to calculate the statistics of the learning process analytically.

We leveraged the 1D map with a symmetric BTSP rule to study the plasticity in a recurrent network, such as in area CA3 of hippocampus, as an animal explores a number of distinct environments. For simplicity we assumed statistically equivalent environments all with a ring topology. This admittedly unrealistic assumption nonetheless allows us to characterize the memory capacity of the network straightforwardly. It seems reasonable to assume that storing memories of environments with different topologies would lead to a similar scaling of memory capacity with system parameters. Indeed, the key determinant of the memory capacity was the sparseness of neuronal coding, i.e. the fraction of place cells in a given environment. For sparse coding, the dynamics of memory recall in the network is equivalent to its projection onto the

low-dimensional manifold corresponding to the desired environment. For the network model we have considered here, this approximation holds already if $s \leq 0.2$. Recent large-scale in-vivo calcium imaging of areas CA1 and CA3 in the hippocampus indicate that the fraction of place cells active in any given environment in both areas is on the order 5% [26], well within the sparse-coding regime of our model. In this sparse coding regime, the interference between memories manifests itself as quenched variability in the connectivity matrix, which is uncorrelated with the environment being analyzed. Perhaps surprisingly, the effect of this variability is to increase the memory capacity by promoting the formation of bump attractors in regions of parameter space which would not otherwise support them. The fact that quenched variability can enhance the robustness of intrinsically generated patterns, as well as increasing the regime of hysteresis near a bifurcation, has been noted in the context of spatially extended systems [27].

In the sparse-coding limit we can therefore approximate the memory capacity of the recurrent network by analyzing the projection of the dynamics on the low-dimensional manifolds corresponding to all past environments. In our case this is equivalent to a series of ring-models, each of which has a connectivity profile which can be calculated analytically from the 1D map. We calculated the memory capacity by determining the age of the environment for which the spatial modulation of the recurrent connectivity was just able to generate a bump attractor, i.e. it stood at a bifurcation to a Turing pattern. In doing so we also took into account the effect of the quenched variability, which always shifted the bifurcation to lower values of the spatial modulation, and hence older memories. It turned out that this calculation was a lower bound on the capacity, as there was generically a significant region of multi-stability already below the bifurcation for which large amplitude bump attractors co-existed with the unpatterned (memoryless) state. All in all, the dynamics of the recurrent neuronal network was well predicted quantitatively by the corresponding ring-model projections as long as the coding was sufficiently sparse. In the limit that the sparseness $s \rightarrow 1$, this approximation broke down, and mixed attractor states formed which were partially correlated with several distinct environments, e.g. see Fig 6.

Our analysis suggests that BTSP, with a temporally symmetric plasticity window, is well suited for one-shot encoding of memories as steady-state attractors in recurrent networks. Such a symmetric form of BTSP has recently been found in recordings from CA3 [28]. It is therefore a plausible candidate mechanism for the formation of episodic memory, a role attributed to the hippocampal formation. For simplicity we have considered the encoding of spatial memory for distinct environments. The global remapping of place cells ensures that such representations will be largely uncorrelated from one environment to another. We have also implicitly assumed that the time between the encoding of one memory and the next is much longer than the time-scale of the plasticity itself. This makes sense for the exploration of distinct environments. Recent theoretical work also suggests that BTSP may be particularly efficient in storing correlated patterns [28]. It nonetheless remains to be determined to what extent BTSP underlies the formation of episodic memory more generally.

## Methods

### Biophysical model (Milstein 2021 [10])

**Summary of the model.**   Here we provide a brief overview of the computational model from [10]. For more details, please refer to that manuscript. They modelled behavioral time scale synaptic plasticity (BTSP) by considering a CA1 place cell which receives inputs from $N$ excitatory CA3 place cells uniformly distributed on a circular track of length $L$. It was assumed that a virtual animal ran at a constant velocity $v$ and, as it crossed a given location denoted by

$x_{PP}$, a plateau potential (PP) occurred either naturally (see [6]) or artificially induced through intracellular current injection.

For each CA3 cell $i$, they modeled the firing rate $R_i$ using a Gaussian function:

$$R_i = R_i(x(t)) = R_{max} \cdot e^{-\frac{1}{2}\left(\frac{y_i - x(t)}{\sigma}\right)^2},$$

where $y_i$ was the peak firing position, the animal's trajectory was $x(t)$ and the parameters $R_{max}$ = 1 and $\sigma = 90/(3\sqrt{2})$. A postsynaptic dendritic PP during each lap $k$ was defined by a binary function:

$$P(x(t)) = \begin{cases} 1 & \text{during a plateau} \\ 0 & \text{otherwise} \end{cases},$$

with a duration of 300 $ms$.

The presynaptic activity of CA3 cells and the PP actived two distinct biochemical signals: an eligibility trace and an instructive signal respectively, and the overlap of the signals drove distinct potentiating and depressing plasticity processes. They modeled these processes using sigmoidal gain functions:

$$q^+(ET_i \cdot IS) = s(ET_i \cdot IS, \alpha^+, \beta^+)$$
$$q^-(ET_i \cdot IS) = s(ET_i \cdot IS, \alpha^-, \beta^-)$$
$$s(x, \alpha, \beta) = \frac{\hat{s}(x, \alpha, \beta) - \hat{s}(0, \alpha, \beta)}{\hat{s}(1, \alpha, \beta) - \hat{s}(0, \alpha, \beta)}$$
$$\hat{s}(x, \alpha, \beta) = \left(1 + e^{-\beta(x-\alpha)}\right)^{-1},$$

where $ET_i$ is the eligibility signal activated by presynaptic neuron $i$ and $IS$ is the instructive signal which propagates to all of the synapses.

The change in weight at each synapse depended on the current value of synaptic weight $w_i$ and the plasticity processes, $q^+$ and $q^-$ with corresponding learning constants $k^+$ and $k^-$:

$$\frac{dw_i}{dt} = (1 - w_i)k^+ q^+(ET_i \cdot IS) - w_i k^- q^-(ET_i \cdot IS), \quad 0 \leq w_i \leq 1.$$

Although the plasticity rule is continuous in time, the total net change in synaptic weight $\Delta w_i$ was computed once per lap integrating the updating from initial time to end time of the track, $t_0$ and $t_1$:

$$\Delta w_i = (1 - w_i)k^+ \Delta Q^+ - w_i k^- \Delta Q^-, \quad 0 \leq w_i \leq 1, \tag{10}$$

where $\Delta Q^* = \int_{t_0}^{t_1} q^*(ET_i \cdot IS)dt$. Parameters of this model are $\tau_{ET}, \tau_{IS}, \alpha^+, \beta^+, \alpha^-, \beta^-, k^+, k^-$.

Parameters for simulations in Fig 1: First, the experimental setup: $L$ = 187 cm, $v$ = 25 cm/s, and $N$ = 100 units. Second, the parameters of the biophysical model: $\tau_{ET}$ = 1664.1 ms, $\tau_{IS}$ = 737 ms, $\alpha^+$ = 0.415, $\beta^+$ = 3.609, $\alpha^-$ = 0.026, $\beta^-$ = 13.815, $k^+$ = 0.9, and $k^-$ = 0.275. And last, the plateau onset position. In Panel F and G, $x_{PP}$ = 93.5 cm, and in panel H there is a sequential induction of PP in two different laps, $x_{PP}^1 = 30$ cm and $x_{PP}^2 = 90$ cm. We plotted the updated weights just after each induction in case of panel H.

## Mathematical modelling: One-dimensional map

**Constructing spatially dependent plasticity functions.** In the biophysical model, both potentiation and depression are asymmetric and skewed with respect to the plateau onset. We

found that both of these processes were well fit by functions proportional to wrapped skew-$t$ distributions.

To do so, we consider $N$ large enough, in limit case, such that we could first define the plasticity rules in time domain using probability density function (PDF) of skew-$t$ distribution [29] with location $\mu$, scale $\sigma$, skewness $\lambda$ and $v$ degrees of freedom:

$$\bar{f}(t) = \frac{2}{\sigma} t_v \left( \frac{t - \mu}{\sigma} \right) T_{v+1} \left( \lambda \frac{t - \mu}{\sigma} \sqrt{\frac{v + 1}{v + \left( \frac{t-\mu}{\sigma} \right)^2}} \right), \tag{11}$$

where $t_v$ and $T_{v+1}$ denote the PDF and cumulative distribution function of the standardised $t$-distribution, and $t = 0$ is the onset time of plateau.

Second, if animal runs at a constant velocity, we can convert time to space $x = v \cdot t + x_{PP}$. Solving this equation by $t$, we have $t = (x - x_{PP})/v$. Replacing $t$ by $(x - x_{PP})/v$ in the Eq 11, the resulting plasticity will be defined in space, $\bar{f}_X(x) = \bar{f}((x - x_{pp}/v))$.

Finally, as animal runs on a linear track with teleportation, as one used in the experiment, the underlying plasticity should be defined on a circle (bounded track), which is equivalent to wrap $\bar{f}_X(x)$ on the circle with length $L$:

$$f(\hat{x}) = \sum_{i \in \mathbb{Z}} \bar{f} \left( \frac{\hat{x} - x_{pp} + i \cdot L}{v} \right), \quad \hat{x} \in [0, L). \tag{12}$$

Converting $\hat{x}$ to $\theta = 2\pi\hat{x}/L$, we have spatial dependent plasticity function for potentiation $f_P(\theta)$ and depression $f_D(\theta)$.

**CA3-CA1 feedforward networks.** Once we defined spatial dependent plasticity functions, we can replace potentiation and depression in Eq 10 by $f_P(\theta)$ and $f_D(\theta)$ respectively. We define $w(\theta)$ the strength of connection from presynaptic CA3 cell that peak at phase $\theta \in [0, \pi)$ to CA1 cell. Therefore, the Eq 10 can be rewrite as

$$\Delta w^k(\theta, \bar{\theta}) = \left( P \cdot (1 - w^{k-1}(\theta)) \cdot f_P(\theta, \bar{\theta}) - D \cdot w^{k-1}(\theta) \cdot f_D(\theta, \bar{\theta}) \right) \cdot \mathbb{I}(\bar{\theta}), \tag{13}$$

where

1. $\bar{\theta}$ is plateau onset phase,

2. $f_P(\theta, \bar{\theta})$ and $f_D(\theta, \bar{\theta})$ are spatial dependent plasticity for potentiation and depression with corresponding learning constants $P$ and $D$ respectively,

3. and $\mathbb{I}(\bar{\theta})$ takes value of 1 if there is plateau potential and 0 otherwise.

Then, the synaptic weights update for each lap $k$, $w^k(\theta)$, according to the 1D map:

$$w^k(\theta) = w^{k-1}(\theta) + \Delta w^k(\theta, \bar{\theta}), \quad w(\theta) \in [0, 1]. \tag{14}$$

And parameters of 1D map are $P, \mu^P, \sigma^P, \lambda^P, v^P, D, \mu^D, \sigma^D, \lambda^D, v^D$.

For the Fig 1, the experimental setup was the same as the biophysical and the model parameters are $P = 2.365$, $\mu^P = 0.685$, $\sigma^P = 1.65$, $\lambda^P = -1.61$, $v^P = 3.5$, $D = 2.57$, $\mu^D = 1.75$, $\sigma^D = 3.65$, $\lambda^D = -5.35$, and $v^D = 5$.

**Recurrent networks.** In this section, we extend the 1D map to recurrent networks, such as in area CA3 of the hippocampus. In this case, we assume that BTSP only shapes the strength of the recurrent excitatory connections between place cells. We therefore assume that cells are already place cells by virtue of spatially-tuned feedforward inputs. The synaptic weight change due to BTSP of a connection from cell $j$ with place field centered at a phase $\theta_j$ to a cell $i$ with

place field centered at a phase $\theta_i$ in an environment $k$, can be written $\Delta w_{ij}(\Delta\theta_{ij}^k)$. The resulting 1D map for the recurrent networks is

$$
\begin{aligned}
w_{ij}^k &= w_{ij}^{k-1} + \Delta w_{ij}(\Delta\theta_{ij}^k), \quad \Delta\theta_{ij}^k = \theta_i^k - \theta_j^k, \\
&= w_{ij}^{k-1} + P(1 - w_{ij}^{k-1})f_P(\Delta\theta_{ij}^k) - Dw_{ij}^{k-1}f_D(\Delta\theta_{ij}^k).
\end{aligned}
\tag{15}
$$

We can leverage the relative simplicity of the map to calculate the memory trace of all past explored environments in the weight matrix. Doing so first requires calculating the mean and variance of the entire weight matrix. Subsequently, to isolate the memory trace of a specific environment $k$, we must first order the neurons according to their phase relationship in that environment. We then determine the mean spatial modulation across the network, e.g. by calculating Fourier coefficients. Finally, we must quantify the quenched variability about this mean connectivity, due to the global remapping of place cells across environments.

**Mean and variance.** After plasticity in a sufficiently large number of environments, the synaptic weight matrix will reach a statistical steady-state. For large learning rates $P$ and $D$ this steady-state is reached quickly, while for small learning rates convergence takes correspondingly longer. Assuming that steady-state has been reached, then the mean weight across the matrix upon exploration of environment $k$ can be written $\langle w_{ij}^k \rangle = \langle w_{ij}^{k-1} \rangle = \mu_w$, where the brackets indicate an average over the matrix without any particular ordering. Applying this average to Eq 15 yields

$$
\mu_w = \frac{P\langle f_P \rangle}{P\langle f_P \rangle + D\langle f_D \rangle}.
$$

Given that the plasticity functions $f_P$ and $f_D$ are defined as periodic in space, in practice, the averages can be taken by ordering the neurons and integrating

$$
\langle f_\alpha \rangle = \langle f_\alpha(\Delta\theta_{ij}^k) \rangle = \frac{1}{2\pi} \int_{-\pi}^{\pi} f_\alpha\left(\Delta\theta_{ij}^k\right) d\left(\Delta\theta_{ij}^k\right), \quad \alpha \in \{P, D\}.
\tag{16}
$$

The steady-state variance $\sigma_w^2 = \langle w^2 \rangle - \langle w \rangle^2$, where we have removed the super- and subscripts from the weights for simplicity. Using Eq 15 we can write

$$
\begin{aligned}
\langle w^2 \rangle &= \langle (w + P(1 - w)f_P - Dwf_D)^2 \rangle \\
&= \langle (Pf_p + wF)^2 \rangle,
\end{aligned}
\tag{17}
$$

where $F = 1 - Pf_p - Df_d$. Solving for $\langle w^2 \rangle$ we find that

$$
\sigma_w^2 = \frac{P^2\langle f_P^2 \rangle + 2P\mu_w\langle f_P F \rangle}{1 - \langle F^2 \rangle} - \mu_w^2
\tag{18}
$$

The horizontal lines in Fig 2D are calculated using Eqs 16 and 18 respectively.

Once a statistical steady-state has been reached, we can calculate the memory trace of past environments. If we consider the weight matrix after an environment $n$, and look back at environment $n - \eta$, it turns out that the statistical properties of the memory trace only depend on the age of the memory, $\eta$, and not the absolute order $n - \eta$. This is true as long as the matrix was already in a steady state after environment $n - \eta - 1$. This point will become clear shortly. In order to quantify the memory trace in environment $n - \eta$, we expand Eq 15 iteratively. Specifically, we write, for the weights after plasticity in environment $n$

$$
w_{ij}^n = Pf_P(\Delta\theta_{ij}^n) + w_{ij}^{n-1}F(\Delta\theta_{ij}^n),
\tag{19}
$$

where we now indicate the ordering of the phases explicitly in each environment. This equation is valid for any $n$, and hence we can expand $w_{ij}^{n-1}$ to express it in terms of $w_{ij}^{n-2}$, and so on. If we do this iteratively until arriving at environment $n - \eta$ we find that

$$w_{ij}^n = \left( P f_P(\Delta\theta_{ij}^{n-\eta}) + w_{ij}^{n-\eta-1} F(\Delta\theta_{ij}^{n-\eta}) \right) \prod_{k=0}^{\eta-1} F(\Delta\theta_{ij}^{n-k}) + P \sum_{l=0}^{\eta-1} f_P(\Delta\theta_{ij}^{n-l}) \prod_{k=0}^{l-1} F(\Delta\theta_{ij}^{n-k}). \tag{20}$$

The memory trace from environment $n - \eta$ is completely contained within the first term. We can see that within the parentheses there is a potentiation term, plus a mixed potentiation and depression term which is premultiplied by the synaptic weight before plasticity. This trace is then degraded in a multiplicative way by all subsequent learning, from environment $n - (\eta - 1)$ to environment $n$. The second term represents an additive noise due to the interference between more recently learned environments, again from $n - (\eta - 1)$ to $n$.

**Amplitude of the memory trace.** We calculate the strength of the memory trace by extracting the first Fourier mode of the spatial modulation in the relevant environment. Here we will assume even plasticity functions $f_P$ and $f_D$ which means that it is sufficient to consider the cosine Fourier component alone. Specifically, if we order the neurons according to their place field location in environment $n - \eta$, we approximate the connectivity between neurons as

$$M_\eta = \mu_w + a_\eta \cdot \cos\left(\Delta\theta^{n-\eta}\right), \tag{21}$$

where $a_\eta$ is the amplitude of the memory trace. In the case of pure cosine plasticity functions, this formula is exact, while for other shapes it can be extended to higher order Fourier modes. Note that because the matrix reached a steady state, this connectivity profile depends only on the age of the memory $\eta$. We determine the amplitude $a_\eta$ through integration

$$\begin{aligned} a_\eta &= \mathbb{R}\left( \langle e^{\vec{i}\Delta\theta_{ij}^{n-\eta}}, w_{ij}^n \rangle \right) \\ &= 2\langle \cos(\Delta\theta_{ij}^{n-\eta}), w_{ij}^n \rangle. \end{aligned}$$

Plugging in the formula for the weight given by Eq 20 and using the fact that the phases in different environments are uncorrelated, i.e. $\forall l \neq n - \eta$

$$\langle e^{\vec{i}\Delta\theta_{ij}^{n-\eta}}, f_P(\Delta\theta_{ij}^l) \rangle = 0,$$

we find that the amplitude is

$$a_\eta = 2\left( P\langle \cos(\Delta\theta_{ij}^{n-\eta}), f_P(\Delta\theta_{ij}^{n-\eta}) \rangle + \mu_w \langle \cos(\Delta\theta_{ij}^{n-\eta}), F(\Delta\theta_{ij}^{n-\eta}) \rangle \right) \langle F \rangle^\eta. \tag{22}$$

The amplitude $a_\eta$ is a mean across the network. That is, if we consider the connectivity profile for a given neuron $i$ after learning, and then average over all $i$, we obtain $a_\eta$ in the limit of many neurons. However, for any given $i$, there will be deviations from this mean in the form of quenched variability. This variability will impact the ability of the network to retrieve a given memory. We can calculate this variability by subtracting the mean connectivity from the weight matrix:

$$V_\eta = \text{Var}(w_{ij}^n - M_\eta) = \langle (w_{ij}^n - a_\eta \cos(\Delta\theta_{ij}^{n-\eta}))^2 \rangle - \mu_w^2. \tag{23}$$

A general expression for $V_\eta$ for arbitrary $f_P$ and $f_D$ could be derived, but would be lengthy. On the other hand, below we provide the explicit formula for a simple choice of $f_P$ and $f_D$ which we use throughout the paper. A more detailed derivation can be found in S1 Appendix.

**Example for simple choice of the plasticity functions $f_P$ and $f_D$.** We choose the simplest possible symmetric rules for potentiation and depression which are roughly consistent with the BTSP rule, namely $f_P(\theta) = 1 + \cos(\theta)$, and $f_D(\theta) = 1 - \cos(\theta)$. With these choices we find the following:

$$\langle f_P \rangle = \langle f_D \rangle = 1, \qquad \langle f_P^2 \rangle = \langle f_D^2 \rangle = \frac{3}{2}, \qquad \langle f_P f_D \rangle = \frac{1}{2},$$

$$\langle w^2 \rangle = \mu(3P^2 - PD - 4P)/(3P^2 + 3D^2 + 2PD - 4P - 4D),$$

$$\langle F \rangle = 1 - P - D,$$

$$\langle F^2 \rangle = 1 + (3P^2 + 3D^2 + 2PD - 4P - 4D)/2.$$

$$\langle \cos(\theta), f_P(\theta) \rangle = \frac{1}{2},$$

$$\langle \cos(\theta), F(\theta) \rangle = \frac{D - P}{2}.$$

With these relations, the mean and variance of the weight matrix are

$$\mu_w = \frac{P}{P + D}, \tag{24}$$

$$\sigma_w^2 = \frac{2P^2 D^2}{(P + D)^2 \cdot (2(PD + P + D) - 3/2(P + D)^2)}, \tag{25}$$

while the mean and variance of the memory trace are

$$a_\eta = \frac{2PD}{P + D}(1 - P - D)^\eta, \tag{26}$$

$$V_\eta = A_\eta + B_\eta \cos(\Delta\theta^{n-\eta}) + C_\eta \cos^2(\Delta\theta^{n-\eta}). \tag{27}$$

The coefficients of the variance curve are

$$\begin{aligned}
A_\eta = \quad & A_0 \langle F^2 \rangle^\eta + \mu^2(\langle F^2 \rangle^\eta - 1) + \frac{3}{2}P^2 \frac{1 - \langle F^2 \rangle^\eta}{1 - \langle F^2 \rangle} \\
& + 2P^2\left(1 - \frac{3}{2}P - \frac{1}{2}D\right)\frac{1}{\langle F \rangle - \langle F^2 \rangle}\left(\frac{1 - \langle F \rangle^\eta}{1 - \langle F \rangle} - \frac{1 - \langle F^2 \rangle^\eta}{1 - \langle F^2 \rangle}\right) \\
& + 2\mu P\left(1 - \frac{3}{2}P - \frac{1}{2}D\right)\frac{\langle F \rangle^\eta - \langle F^2 \rangle^\eta}{\langle F \rangle - \langle F^2 \rangle},
\end{aligned}$$

$$\begin{aligned}
B_\eta = \quad & B_0 \langle F^2 \rangle^\eta + 2a_0\mu(\langle F^2 \rangle^\eta - \langle F \rangle^{2\eta}) \\
& + 2a_0 P\left(\left(1 - \frac{3}{2}P - \frac{1}{2}D\right)\frac{\langle F \rangle^\eta - \langle F^2 \rangle^\eta}{\langle F \rangle - \langle F^2 \rangle} - \frac{\langle F \rangle^\eta - \langle F \rangle^{2\eta}}{1 - \langle F \rangle}\right),
\end{aligned}$$

$$C_\eta = \quad C_0 \langle F^2 \rangle^\eta + a_0^2\left(\langle F^2 \rangle^\eta - \langle F \rangle^{2\eta}\right).$$

where

$$A_0 = P^2 + 2P(1-P-D)\mu + (1-P-D)^2\langle w^2\rangle - \mu^2,$$

$$B_0 = 2\frac{P-D}{P+D}\left(P^2 + P(1-2P-2D)\mu - (P+D)(1-P-D)\langle w^2\rangle\right),$$

$$C_0 = \left(\frac{P-D}{P+D}\right)^2\left(P^2 - 2P(P+D)\mu + (P+D)^2\langle w^2\rangle\right).$$

(28)

**Taking into account network size and sparseness.** In the previous section we did not explicitly specify the system size. However, the continuum approximation for the spatial plasticity functions implicitly assumes that there are place cells distributed around the circle with sufficient spatial resolution. The number of spatial positions, $N$, therefore should not be too small, but beyond this does not explicitly enter into our calculations of the network connectivity. On the other hand, the mean and variance of the memory traces have been calculated for precisely one cell at each position. Namely, $w_{ij}$ represents the connection from a single cell $j$ to a single cell $i$.

We now introduce network size explicitly by considering a number of cells at each position. Specifically, we assume that there are $M$ place cells with place fields at each of the $N$ positions, and that only a fraction $s$ of those cells are active in any given environment. Importantly, the global remapping from one environment to the next is of each individual cell, not each group of $M$ cells. That is, the identify of the cells in each group of $M$ cells changes from environment to environment.We can take the sparseness into account in the 1D map by keeping track of which neurons are active in any given environment, and hence which synapses get updated. The rule is now

$$w_{ij}^n = w_{ij}^{n-1} + \left(P\cdot\left(1-w_{ij}^{n-1}\right)f_P\left(\Delta\theta_{ij}^n\right) - D\cdot w_{ij}^{n-1}\cdot f_D\left(\Delta\theta_{ij}^n\right)\right)S_i^n S_j^n,$$

(29)

where 1) $i$ and $j$ goes from 1 to $MN$ and 2) $S_i^n$ takes value of 1 or 0, active or not in environment $n$. We assume the activation of neurons is a Bernoulli process with probability $s$, i.e., $P(S_i^n = 1) = s$. Using this map, we can conduct the same analysis as before.

For the choice of

$$f_P(\theta) = 1 + \cos(\theta), \quad f_D(\theta) = 1 - \cos(\theta),$$

the mean and variance of the weight matrix, $\mu_w$ and $\sigma_w^2$ remain the same, see Eqs 24–25. Rewriting the weight equation as function of environment $n - \eta$ to compute the mean and variance of memory trace in sparse network gives:

$$w_{ij}^n = \left(Pf_P(\theta_{ij}^{n-\eta})S_i^{n-\eta}S_j^{n-\eta} + w_{ij}^{n-\eta-1}F(\theta_{ij}^{n-\eta})\right)\prod_{k=0}^{\eta-1}F(\Delta\theta_{ij}^{n-k})$$

$$+ P\sum_{l=0}^{\eta-1}f_P(\theta_{ij}^{n-k})S_i^{n-k}S_j^{n-k}\prod_{k=0}^{l-1}F(\Delta\theta_{ij}^{n-k})$$

(30)

where $F(\Delta\theta_{ij}^k) = 1 - S_i^k S_j^k(Pf_P(\Delta\theta_{ij}^k) + Df_D(\Delta\theta_{ij}^k))$. Since the probability of activation of cells is

uncorrelated, we have

$$\langle F \rangle \quad = 1 - s^2(P + D) \quad \text{and}$$

$$\langle F^2 \rangle \quad = 1 + s^2(3P^2 + 3D^2 + 2PD - 4P - 4D)/2,$$

where $\langle (S_i^k)^2, (S_j^k)^2 \rangle = \langle S_i^k, S_j^k \rangle = \langle S_i^k \rangle \langle S_j^k \rangle = s^2$. When we calculate the mean and variance of the memory trace in environment $n - \eta$, we only consider the subset of active cells, and hence take $S_i^{n-\eta} = 1$, while for all other environments $k$, $S_i^k$ is treated as a Bernoulli random variable. We calculate the mean and variance of the memory trace as before, see Supplementary Materials for detailed calculations, yielding for the mean

$$a_\eta = 2\langle f_1^{n-\eta} \rangle = \left( \frac{2PD}{P + D} \right)(1 - s^2(P + D))^\eta. \tag{31}$$

The variance is

$$V_\eta \quad = A_\eta + B_\eta \cos(\Delta\theta^{n-\eta}) + C_\eta \cos^2(\Delta\theta^{n-\eta}), \tag{32}$$

where

$$A_\eta = \quad A_0 \langle F^2 \rangle^\eta + \mu^2(\langle F^2 \rangle^\eta - 1) + \frac{3}{2}P^2 s^2 \frac{1 - \langle F^2 \rangle^\eta}{1 - \langle F^2 \rangle}$$

$$+ 2P^2 s^4 \left( 1 - \frac{3}{2}P - \frac{1}{2}D \right) \frac{1}{\langle F \rangle - \langle F^2 \rangle} \left( \frac{1 - \langle F \rangle^\eta}{1 - \langle F \rangle} - \frac{1 - \langle F^2 \rangle^\eta}{1 - \langle F^2 \rangle} \right)$$

$$+ 2\mu P s^2 \left( 1 - \frac{3}{2}P - \frac{1}{2}D \right) \frac{\langle F \rangle^\eta - \langle F^2 \rangle^\eta}{\langle F \rangle - \langle F^2 \rangle},$$

$$B_\eta = \quad B_0 \langle F^2 \rangle^\eta + 2a_0 \mu \left( \langle F^2 \rangle^\eta - \langle F \rangle^{2\eta} \right)$$

$$+ 2a_0 P s^2 \left( \left( 1 - \frac{3}{2}P - \frac{1}{2}D \right) \frac{\langle F \rangle^\eta - \langle F^2 \rangle^\eta}{\langle F \rangle - \langle F^2 \rangle} - \frac{\langle F \rangle^\eta - \langle F \rangle^{2\eta}}{1 - \langle F \rangle} \right),$$

$$C_\eta \quad = C_0 \langle F^2 \rangle^\eta + a_0^2 \left( \langle F^2 \rangle^\eta - \langle F \rangle^{2\eta} \right).$$

and $A_0$, $B_0$, and $C_0$ remain the same, see Eq 28. For a detailed derivation of Eqs 31 and 32 see S1 Appendix.

## Calculating the SNR from the synaptic weight matrix

Once we have computed the mean and variance of memory trace, we can use the signal-to-noise ratio (SNR) to approximate the storage capacity in a purely structural sense. We assume that a memory can be retrieved if the SNR is greater than some threshold value, for example $T = 1$. Then the storage capacity is

$$\eta_{max} = \max_\eta\{\text{SNR}_\eta >= 1\} = \max_\eta\left\{\frac{a_\eta}{\sqrt{\langle V_\eta \rangle}} >= 1\right\}.$$

In the general case, the variance is itself a function of space, making comparison with the amplitude $a_\eta$ complicated. Here we consider only the component of the variance which is

constant in space, and additionally with $P = D$, for which the formulas simplify to

$$
\begin{aligned}
a_\eta &= P(1 - 2s^2P)^\eta \\
A_\eta &= \frac{P}{8(1-P)} - \frac{P^2}{2}(1 - 4s^2P + 4s^2P^2)^\eta \quad \text{and} \\
C_\eta &= P(1 - 4s^2P + 4s^2P^2)^\eta - P^2(1 - 2s^2P)^{2\eta}.
\end{aligned}
$$

Then, the SNR for the environment $\eta$ is

$$
SNR_\eta = \frac{a_\eta}{\sqrt{\langle V_\eta \rangle}} = \frac{a_\eta}{\sqrt{\left(a_\eta + \frac{1}{2}C_\eta\right)/sM}} = \frac{\sqrt{sM}P(1 - 2s^2P)^\eta}{\sqrt{P/(8 - 8P) - P^2(1 - 2s^2P)^{2\eta}/2}} \tag{33}
$$

and the memory capacity is

$$
\eta_{max} = -\Big(\ln 8(1 - P)(sM + 1/2)\Big)/(2\ln(1 - 2s^2P)). \tag{34}
$$

If we assume that $s^2 \ll 1$ and that $sM \gg 1/2$, which is the case if we wish to obtain a large capacity, then the formula further simplifies to

$$
\eta_{max} = \frac{1}{4s^2P}\ln\Big(sM \cdot 8P(1 - P)\Big). \tag{35}
$$

Taking only the contribution of the system size $sM$ in the logarithm leads to Eq 6.

## Network model

The 1D map for BTSP allows us to calculate the strength of the memory trace corresponding to all past environments in the connectivity matrix itself. Having a sufficiently strong SNR of the memry trace is a necessary, but not sufficient condition for memory recall. In order to study the dynamics of memory recall, we consider a network of rate neurons with recurrent excitatory connectivity given by the weight matrix resulting from the plasticity rule. For simplicity, we model the effect of inhibition as being proportional to the total excitatory activity. Such an approximation would result, for example, if the dynamics of inhibitory interneurons were much faster than that of excitatory neurons. We also take into account the fact that the plasticity rule considers normalized weight, and therefore rescale the weights by an overall maximum weight factor. Specifically, we define the rescaled synaptic weight used in the network model from a neuron $j$ to a neuron $i$ as $\bar{w}_{ij} = W_0 + W_{max}(w_{ij}^n - \mu_w)$, where $w_{ij}^n$ is the weight from the plasticity rule after exploration of environment $n$, and $\mu_w = \langle w_{ij} \rangle$ is the mean weight. The plasticity rule assumes that there are place cells with place fields located at $N$ uniformly distributed positions in any given environment. At each position there are $M$ cells, but only a fraction $s$ of them are active. The total number of neurons in the network is therefore $MN$, while the number of active neurons in any given environment is $sMN$.

The firing rate equations are

$$
\begin{cases}
\tau \dfrac{d}{dt} r_1 = & -r_1 + \phi\left( \dfrac{1}{\kappa N} \sum\limits_{j=1}^{MN} \bar{w}_{1j} r_j S_j^k + I_0 \right) S_1^k \\[2ex]
\tau \dfrac{d}{dt} r_2 = & -r_2 + \phi\left( \dfrac{1}{\kappa N} \sum\limits_{j=1}^{MN} \bar{w}_{2j} r_j S_j^k + I_0 \right) S_2^k \\[2ex]
\quad \vdots & \\[1ex]
\tau \dfrac{d}{dt} r_i = & -r_i + \phi\left( \dfrac{1}{\kappa N} \sum\limits_{j=1}^{MN} \bar{w}_{ij} r_j S_j^k + I_0 \right) S_i^k \\[2ex]
\quad \vdots &
\end{cases}
\tag{36}
$$

where $r_i$ is the firing rate of cell $i$, $\phi$ is a nonlinear fI curve, and $S_i^k = 1$ is cell $i$ is active in environment $k$ and otherwise is 0. For simplicity we assume that the external drive $I_0$ is the same for all neurons. In simulations we take $\phi(x) = 0$ if $x < 0$, $x^2$ if $0 \leq x \leq 1$ and $2\sqrt{x - 3/4}$ if $x > 1$. This function is continuous and smooth and captures the expansive nonlinearity at low rates typical for irregularly firing neurons in the fluctuation-driven regime. The parameter $\kappa$ sets how we scale the connectivity with respect to the size of the neuronal population encoding each position. As we shall see in the following section, taking $\kappa = \sqrt{sM}$ allows us to compare the memory capacity directly with the SNR calculation. With this scaling, the mean input grows with population size, while the variance remains order one. Alternatively we can take $\kappa = sM$ for which the mean remains constant but the variances decreases with increasing population size.

## Ring-model approximation

The connectivity in the network model is correlated with all past environments. When we simulate the dynamics in a given environment $k$, we set $S_i^k = 1$ for all neurons with place fields in that environment, and otherwise set $S_i^k = 0$. We then typically order the neurons according to their preferred place field in environment $k$ and consider an initial condition of either a small-amplitude, or a large-amplitude spatial modulation with that same ordering. We can consider an approximation to the full network dynamics which is restricted to the manifold corresponding to spatial modulations exclusively in environment $k$. Such an approximation is equivalent to a ring-model for this environment. Note that there is no guarantee that the full dynamics will actually be confined to this manifold. In practice, we find that for sparse enough neuronal representations (small $s$), the ring-model approximation works well, while as $s \rightarrow 1$ it breaks down and mixed attractor states emerge which cannot be captured in a ring-model framework.

The ring model for environment $n - \eta$ is:

$$
\tau \frac{\partial}{\partial t} r(\theta^{n-\eta}, t) = -r(\theta^{n-\eta}, t) + \phi\left( \frac{1}{2\pi} \int_{-\pi}^{\pi} W(\theta^{n-\eta} - \theta') r(\theta', t) d\theta' + I_0 \right),
\tag{37}
$$

where we have assumed $N \gg 1$. Importantly, we choose the connectivity so as to match that from the full network model. Specifically, we take

$$
W(\Delta\theta^{n-\eta}) = W_0 + W_1^\eta \cos\left( \Delta\theta^{n-\eta} \right) + \Delta W z(\Delta\theta^{n-\eta}),
$$

where $W_1^\eta = W_{max} \frac{sM}{\kappa} a_\eta$, $\Delta W = W_{max} \frac{\sqrt{sM}}{\kappa} \sqrt{V_\eta}$, and $z$ a zero-mean Gaussian random variable

with unit variance. Therefore, we have a set of ring models, one for each environment:

$$
\begin{cases}
\tau \dfrac{\partial}{\partial t} r(\theta^n, t) = & -r(\theta^n, t) + \phi\left(\dfrac{1}{2\pi}\displaystyle\int_{-\pi}^{\pi} W(\theta^n - \theta')r(\theta', t)d\theta' + I_0\right) \\[2ex]
\tau \dfrac{\partial}{\partial t} r(\theta^{n-1}, t) = & -r(\theta^{n-1}, t) + \phi\left(\dfrac{1}{2\pi}\displaystyle\int_{-\pi}^{\pi} W(\theta^{n-1} - \theta')r(\theta', t)d\theta' + I_0\right). \\[2ex]
\tau \dfrac{\partial}{\partial t} r(\theta^{n-2}, t) = & -r(\theta^{n-2}, t) + \phi\left(\dfrac{1}{2\pi}\displaystyle\int_{-\pi}^{\pi} W(\theta^{n-2} - \theta')r(\theta', t)d\theta' + I_0\right) \\[2ex]
& \vdots
\end{cases}
$$

Note that in the limit of the ring-model approximation, the interaction between different environments is expressed by the quenched variability $\Delta W$.

**Linear stability analysis and Turing bifurcation.** Stationary uniform solutions (SU) of Eq 37 are given by

$$
r_0 = \phi(W_0 r_0 + I_0),
$$

where $r_0$ is a constant, and spatially uniform firing rate. In order to determine if bump attractors can emerge spontaneously, we calculate the linear stability of the spatially uniform state.

We write the ansatz

$$
r(\theta, t) = r_0 + \delta r_0 e^{\lambda_0 t} + \delta r_1 \cos(\theta) e^{\lambda_1 t},
$$

where $\delta r \ll 1$.

**Without quenched variability.** If the number of place cells encoding each position along the circular track is very large, we can ignore the quenched variability, i.e. $\Delta W \to 0$. Then, plugging the ansatz for the small amplitude perturbation into Eq 37, expanding and collecting terms results in the two eigenvalues, $\lambda_0 = -1 + \phi_0' W_0$ and $\lambda_1 = -1 + \phi_0' \frac{W_1}{2}$, where $\phi_0'$ is the slope of the transfer function evaluated at the steady state. To avoid a uniform instability of the steady state $W_0$ must be chosen $W_0 < 1/\phi_0'$. Then we find that the critical value of the spatially modulated connectivity

$$
W_1^{cr} = \frac{2}{\phi_0'}. \tag{38}
$$

If we substitute the value of $W_1$ for environment $n - \eta$ into this formula, we find that

$$
\eta_{cr} = \frac{1}{s^2(P+D)} \ln\left(\frac{sM}{\kappa} W_{max} \phi_0' \frac{PD}{(P+D)}\right). \tag{39}
$$

Taking $P = D$ and $\kappa = \sqrt{sM}$, separating the logarithm and taking the leading order term for large $sM$ leads to Eq 6.

**With quenched variability.** In order to take the effect of quenched variability on the Turing bifurcation into account, we express the Gaussian noise process $\Delta W(\theta) = \sqrt{V(\theta)}z$ in terms of its Fourier series,

$$
\begin{aligned}
\Delta W(\theta) &= \sum_{j=1}^{N}(c_j e^{ij\theta} + \bar{c}_j e^{-ij\theta}), \\
&= 2\sum_{j=1}^{N}\alpha_j \cos(j\theta) + 2\sum_{j=1}^{N}\beta_j \sin(j\theta),
\end{aligned} \tag{40}
$$

where $N$ is the number of positions, equal to the number of populations of place cells. Because the variability is a zero-mean Gaussian process, the coefficients $\alpha_j$ and $\beta_j$ are also zero-mean Gaussian random variables whose variances and covariances must be determined self-consistently. We do this through appropriate averaging. Specifically, we calculate $V(\theta) = \langle \Delta W(\theta)^2 \rangle$, which yields

$$
\begin{aligned}
V(\theta) \quad = \quad & 2\sum_{j=1}^{N}\sum_{l=1}^{N}\Big(\langle\alpha_j\alpha_l\rangle + \langle\beta_j\beta_l\rangle\Big)\cos((j-l)\theta)) \\
& +2\sum_{j=1}^{N}\sum_{l=1}^{N}\Big(\langle\alpha_j\alpha_l\rangle - \langle\beta_j\beta_l\rangle\Big)\cos((j+l)\theta)) \\
& +4\sum_{j=1}^{N}\sum_{l=1}^{N}\Big(\langle\alpha_j\beta_l\rangle\sin((j-l)\theta) - \langle\alpha_j\beta_l\rangle\sin((j+l)\theta)\Big).
\end{aligned}
\tag{41}
$$

We recall that $V(\theta) = A + B\cos(\theta) + C\cos(\theta)^2 = A + \frac{C}{2} + B\cos(\theta) + \frac{C}{2}\cos(2\theta)$. Now we determine the statistics of the coefficients in Eq 41 by applying Parseval's Theorem, and assuming that the power is distributed evenly amongst all N terms on the right hand side, as is the case for a white noise process. Doing so yields

$$
\langle\alpha_1^2\rangle \quad = \quad \frac{1}{2N}\left(A + \frac{3C}{4}\right),
\tag{42}
$$

$$
\langle\beta_1^2\rangle \quad = \quad \frac{1}{2N}\left(A + \frac{C}{4}\right),
\tag{43}
$$

$$
\langle\alpha_{j\neq1}^2\rangle = \langle\beta_{j\neq1}^2\rangle \quad = \quad \frac{1}{2N}\left(A + \frac{C}{2}\right),
\tag{44}
$$

$$
\langle\alpha_j\alpha_{j+1}\rangle = \langle\beta_j\beta_{j+1}\rangle \quad = \quad \frac{B}{4N},
\tag{45}
$$

$$
\langle\alpha_j\alpha_{j+2}\rangle = \langle\beta_j\beta_{j+2}\rangle \quad = \quad \frac{C}{8N},
\tag{46}
$$

see S2 Appendix for a detailed derivation. Eq 44 shows that for finite $N$ the quenched variability generates power in the relevant mode for a bump instability ($j = 1$). The amplitude of the modulation is given by $R_1 = 2\sqrt{\alpha_1^2 + \beta_1^2}$. Because $\alpha$ and $\beta$ are random variables, the amplitude (and the phase) of the modulation for each cell will also be a random variable. The mean of R can be found by integrating over the distributions of $\alpha$ abd $\beta$, yielding

$\langle R \rangle = \sqrt{\frac{2\pi}{N}\left(A + \frac{C}{2}\right)} \cdot \frac{\sqrt{sM}}{\kappa}$. Finally, we find that the critical value of the connectivity is now

$$
W_1^{cr} = \frac{2}{\phi_0'} - \langle R \rangle,
\tag{47}
$$

which indicates that the bifurcation occurs for lower values of $W_1$ than the case without quenched variability.

## Supporting information

**S1 Appendix. Detailed calculation of memory traces and quenched variability in the BTSP rule.**
(PDF)

**S2 Appendix. Calculation of the Spatial Fourier Spectrum for Quenched White Noise.**
(PDF)

**S1 Fig. Comparison of map with biophysical model for different running velocities and induction protocols. (A)** Potentiation (top) and depression (bottom) plasticity functions for different velocities. **(B-C)** $v = 25cm/s$. **(B)** Synaptic weights after a single induction (left) and after five inductions at the same position on consecutive trials (right). **(C)** The results of two inductions at distinct locations on the same trial, and the subsequent change in weights after a third induction on the second trial (right). **(D-E)** The 1D map quantitatively captures the degree of plasticity inferred from the biophysical model with different parameter set. **Parameters:** (A) **Velocity = 20**: $P = 2.625$, $\mu^P = 0.75$, $\sigma^P = 1.825$, $\lambda^P = -1.55$, $v^P = 5$, $D = 2.8$, $\mu^D = 1.875$, $\sigma^D = 3.45$, $\lambda^D = -4.55$, and $v^D = 5$; **velocity = 35**: $P = 2$, $\mu^P = 0.6$, $\sigma^P = 1.42$, $\lambda^P = -1.625$, $v^P = 2$, $D = 2.15$, $\mu^D = 1.54$, $\sigma^D = 3.85$, $\lambda^D = -5.75$, and $v^D = 3$. (B-C) Model parameters are the same as Fig 1. In panel B, $x_{pp}^i = 93.5$ cm, middle of the track, for $i \in \{1, 2, 3, 4, 5\}$. In panel C, $x_{pp}^1 = 30, 120$ cm and $x_{pp}^2 = 93.5$ cm. (D-E) Setup of simulation: $v = 35cm/s$, $L = 187cm$, $x_{pp} = 93.7$ cm. Parameters of biophysical model for panel D (E) are: $\tau_{Elig} = 0.98(0.8)$, $\tau_{IS} = 0.64(0.5)$, $\alpha_+ = 0.19(0.2)$, $\beta_+ = 23.82(30)$, $\alpha_- = 0.05(0.1)$, $\beta_- = 730.64(1000)$, $k_+ = 0.8(0.8)$, $k_- = 0.44(0.4)$. The corresponding parameters of 1D map are $v_P = 6(5)$, $\mu_P = 0.95(0.8)$, $\sigma_P = 1.45(1.35)$, $\lambda_P = -4.5$ $(-6)$, $P = 2(1.6)$, $v_D = 5(5)$, $\mu_D = 1.6(1.1)$, $\sigma_D = 2.5(1.6)$, $\lambda_D = -6(-7)$, $D = 2.65(1.2)$.
(TIF)

**S2 Fig. The value of $P$ and $D$ can change the shape of spatial statistics. (A-C)** a. Global statistics, average (top) and variance (bottom), as function of number of explored environments. b. Spatial statistics of weight matrix at steady state: mean curve with 95% confidence interval (top) and variance of noise (bottom) as function of phase difference. **Parameters:** (A) $P = D = 0.1$; (B) $P = 0.1$ and $D = 0.3$; (C) $P = 0.3$ and $D = 0.1$.
(TIF)

**S3 Fig. Signal-to-noise ratio and storage capacity for different values of $P$ and $D$. (A-C)** a. Signal-to-noise ratio as function of number of past explored environments by fixing either sparsity (top) or network size (bottom). Dashed line indicates the inferior threshold that no memory trace lower than this value. b. Memory capacity of network as function of network size (top) or sparsity (bottom). **(D-E)** $P = D$. **(D)** The memory capacity of system as function of $P$. **(E)** The optimal value of $P$ that maximize the capacity of system as function of population size $M$.
(TIF)

**S4 Fig. Signal-to-noise ratio and storage capacity for different values of $P$ and $D$. (A)** Left top: Zoomed plot of the Fig 4C showing bump amplitude in individual fashion. Right top: Detail of diagram highlighting the emergence of a small amplitude, supercritical Turing bifurcation. Bottom: sample profile of large and small bump amplitude. **(B-D)**: Phase diagrams of the ring model without noise. The solid line indicates the analytical expression of the Turing bifurcation, while the dashed line indicates the saddle-node of bump solution. Red dashed line represents the same network setup across phase diagrams.
(TIF)

**S5 Fig. Same as in Fig 7 of the main text but with $s = 0.2$.**
(TIF)

## Author Contributions

**Conceptualization:** Pan Ye Li, Alex Roxin.

**Data curation:** Pan Ye Li.

**Formal analysis:** Pan Ye Li, Alex Roxin.

**Funding acquisition:** Alex Roxin.

**Investigation:** Pan Ye Li, Alex Roxin.

**Methodology:** Pan Ye Li, Alex Roxin.

**Project administration:** Pan Ye Li.

**Software:** Pan Ye Li.

**Supervision:** Alex Roxin.

**Visualization:** Pan Ye Li, Alex Roxin.

**Writing – original draft:** Pan Ye Li, Alex Roxin.

**Writing – review & editing:** Pan Ye Li, Alex Roxin.

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
