## [Decision Letter · Decision Letter 0]

30 May 2023

Dear Dr. Roxin,

Thank you very much for submitting your manuscript "Rapid memory encoding in a recurrent network model with behavioral time scale synaptic plasticity" for consideration at PLOS Computational Biology.

As with all papers reviewed by the journal, your manuscript was reviewed by members of the editorial board and by several independent reviewers. In light of the reviews (below this email), we would like to invite the resubmission of a significantly-revised version that takes into account the reviewers' comments.

In particular, it is important for the authors to clarify how the parameters of their BTSP rule correspond to the established experimental data, as highlighted by both reviewers. It is also crucial that any model of this type makes testable predictions that render it falsifiable - and some comparison with previous synaptic plasticity models that have attempted to account for the same hippocampal memory phenomena would be useful, in this regard.

We cannot make any decision about publication until we have seen the revised manuscript and your response to the reviewers' comments. Your revised manuscript is also likely to be sent to reviewers for further evaluation.

Sincerely,

Daniel Bush

Academic Editor

PLOS Computational Biology

Thomas Serre

Section Editor

PLOS Computational Biology

Reviewer's Responses to Questions

**Comments to the Authors:**

Reviewer #1: The review is uploaded as an attachment.

Reviewer #2: Summary:

In this study, the authors analyze a simplified 1D map derived from experimental BTSP rules. They demonstrate that this simplified map aligns with biophysical models (i.e., experimental BTSP rule) and investigate the properties of memory storage in recurrent neural networks, such as CA3. They studied the correlation of the synaptic weight matrix created by the rules and estimate the properties of memory capacity through signal-to-noise ratio (SNR) analysis. The authors also provide good analysis for the sparse coding and the properties of attractors.

Overall, this work provides valuable insights into the memory storage of simplified BTSP in recurrent neural networks, specifically CA3. The simplified map captures the essential features of BTSP, and the analysis of recurrent neural networks enables to leverage the SNR tools for studying memory capacity.

Comments:

The assumption that each cell induces plateau potentials may oversimplify the sparsity and randomness of PPs. Please provide justifications, references or some discussion for this assumption.

Please provoide more details on how BTSP constructs the connection weights and how it operates in firing rate models when simulating the network of firing rate neurons.

Although the simplified rule aligns with the biophysical model, it is worth noting that the parameters used for simulations fall outside the range of experimental fitting (Milstein 2021 [10]). In fact, the heapmap of the change in weights shown in Fig.1G differs from that of (see Fig.3I of Milstein 2021). Considering that the biophysical model allows for nine free parameters, it would be beneficial to demonstrate that the model accurately fits the experimental curve suggested by previous studies.

Since BTSP is primarily discovered in the connection between CA3 and CA1, it would be valuable if the authors could discuss the applicability of the proposed rule to CA3 as well.

**Have the authors made all data and (if applicable) computational code underlying the findings in their manuscript fully available?**

Reviewer #1: **No: **Data availability states that code will be made available upon publicaiton, but it was not available at time of submission

Reviewer #2: Yes

PLOS authors have the option to publish the peer review history of their article (what does this mean?). If published, this will include your full peer review and any attached files.

Reviewer #1: No

Reviewer #2: No
---

## [Decision Letter · Decision Letter 1]

10 Jul 2023

Dear Dr. Roxin,

We are pleased to inform you that your manuscript 'Rapid memory encoding in a recurrent network model with behavioral time scale synaptic plasticity' has been provisionally accepted for publication in PLOS Computational Biology.

Best regards,

Daniel Bush

Academic Editor

PLOS Computational Biology

Thomas Serre

Section Editor

PLOS Computational Biology

Reviewer's Responses to Questions

**Comments to the Authors:**

Reviewer #1: A recent preprint from the lab of Jeff Magee which found symmetric BTSP kernels in CA3 (Li et al. 2023) assuaged my major concern for the paper. All prior BTSP publications had noted an asymmetric kernel, so I had felt it was important for models of BTSP to include this component. However, with this new finding, the symmetric kernel in the model is clearly justified, and I am now happy to recommend the work. My other concerns were minor and mostly addressed by the latest revision. Thank you to the authors for their detailed responses to my comments.

Reviewer #2: Thanks for the authors' response, which has addressed my concerns. I am glad to see that the symmetric assumption made in this work is in line with the recent experimental model in CA3 from Maggee's lab. Consequently, I endorse to the publication of this paper.

**Have the authors made all data and (if applicable) computational code underlying the findings in their manuscript fully available?**

Reviewer #1: Yes

Reviewer #2: Yes

PLOS authors have the option to publish the peer review history of their article (what does this mean?). If published, this will include your full peer review and any attached files.

Reviewer #1: No

Reviewer #2: **Yes: **Yujie Wu

---

## [Editor Report · Acceptance letter]

1 Aug 2023

PCOMPBIOL-D-23-00687R1 

Rapid memory encoding in a recurrent network model with behavioral time scale synaptic plasticity

Dear Dr Roxin,

I am pleased to inform you that your manuscript has been formally accepted for publication in PLOS Computational Biology. Your manuscript is now with our production department and you will be notified of the publication date in due course.

With kind regards,

Zsofi Zombor
